# Tectonic and climatic drivers of Asian monsoon evolution

James R. Thomson [1], Philip B. Holden [2✉], Pallavi Anand [2], Neil R. Edwards [2,3], Cécile A. Porchier [2,4] & Nigel B. W. Harris [2]

Asian Monsoon rainfall supports the livelihood of billions of people, yet the relative importance of different drivers remains an issue of great debate. Here, we present 30 million-year model-based reconstructions of Indian summer monsoon and South East Asian monsoon rainfall at millennial resolution. We show that precession is the dominant direct driver of orbital variability, although variability on obliquity timescales is driven through the ice sheets. Orographic development dominated the evolution of the South East Asian monsoon, but Indian summer monsoon evolution involved a complex mix of contributions from orography (39%), precession (25%), atmospheric $CO_2$ (21%), ice-sheet state (5%) and ocean gateways (5%). Prior to 15 Ma, the Indian summer monsoon was broadly stable, albeit with substantial orbital variability. From 15 Ma to 5 Ma, strengthening was driven by a combination of orography and glaciation, while closure of the Panama gateway provided the prerequisite for the modern Indian summer monsoon state through a strengthened Atlantic meridional overturning circulation.

[1] Safety in Engineering Ltd, South Lanarkshire, UK. [2] School of Environment, Earth & Ecosystem Sciences, The Open University, Milton Keynes, UK. [3] Cambridge Centre for Energy, Environment and Natural Resource Governance, University of Cambridge, Cambridge, UK. [4] Present address: Department of Geography, University College London, London, UK. ✉email: Philip.Holden@open.ac.uk

Proxy reconstructions and modelling studies have shown multiple drivers of Asian monsoon variability[1]: orbital forcing, atmospheric carbon dioxide ($CO_2$), global ice volume, Himalayan-Tibetan Plateau (HTP) uplift, tectonically-induced changes to major ocean gateways, and Intertropical Convergence Zone (ITCZ) movement, via atmospheric circulation changes and Atlantic Meridional Overturning Circulation (AMOC) teleconnections. However, a comprehensive and systematic assessment of all these drivers together has been lacking but is necessary to quantify their relative roles and the interactions between them. A recent modelling study[2] based on snapshot HadCM3 simulations at 3 million-year (Myr) intervals, for the past 150 Myr, suggested that tectonic uplift dominated the evolution of the East Asian Monsoon, one of the subsystems of the Asian Monsoon[2]. However, this approach does not adequately address the relative effects of different drivers against one another, or the interactions between multiple drivers, noting that these drivers have been shown to impact monsoon dynamics over a range of timescales[3].

On the timescales of interest, transient simulations of sufficient length to be useful are computationally intractable; even the most efficient atmosphere-ocean GCMs would require thousands of years of computing. We overcome this problem by using emulation, a widely used approach in climate science that generates fast approximations to simulations by deriving statistical relationships between model inputs and outputs. We build emulators of monsoon rainfall simulated by the intermediate complexity Earth system model PLASIM-GENIE[4] in response to changing boundary conditions for the last 30 Myr. We consider the Asian Monsoon subsystems (ISM and SEAM) separately as they are known to be influenced by distinctly different wind systems and moisture sources[1,5]. This fact has incited considerable debate

about their coupling on orbital and millennial scales[6–9]. We use these emulators to perform a global sensitivity analysis[10], enabling us to disentangle the roles of all the principal drivers on the ISM and SEAM, and then to generate time-series of rainfall reconstructions at high (1000-year) resolution by providing the proxy-based boundary conditions and timing of major tectonic changes as inputs to the emulators (Figs. 1 and 2). By simulating three alternative paleogeographies, we capture the first-order effects of the two main ocean gateways, but not the full effects of the retreat of the Tethys and Paratethys oceans[11–13], still ongoing in the Miocene[14], with increasing continentality displacing pressure systems and triggering changes in monsoon strength.

In this work, we show that while long-term SEAM evolution is dominated by orographic development, which drives 93% of the modelled SEAM variance since 30 Ma, controls on the strength of the ISM are more complex, and include an important role for atmospheric $CO_2$ (driving 21% of the modelled ISM variance) and significant roles for ice-sheets and ocean gateways. The orbital variability of both systems is dominated by precession, although variability over obliquity time scales is driven through ice sheets.

## Results and discussion

**Sensitivity to climate drivers.** Summer rainfall dominates the modern Indian and South East Asian monsoons as winter monsoons are dry. We emulate annual precipitation, which we regard as the variable most representative of precipitation and runoff proxies, and apply the terms ISM and SEAM, noting that the area covered in the latter is commonly referred to as East Asian Summer Monsoon (EASM). The modelled areas of the ISM and SEAM are shown in Fig. 3a.

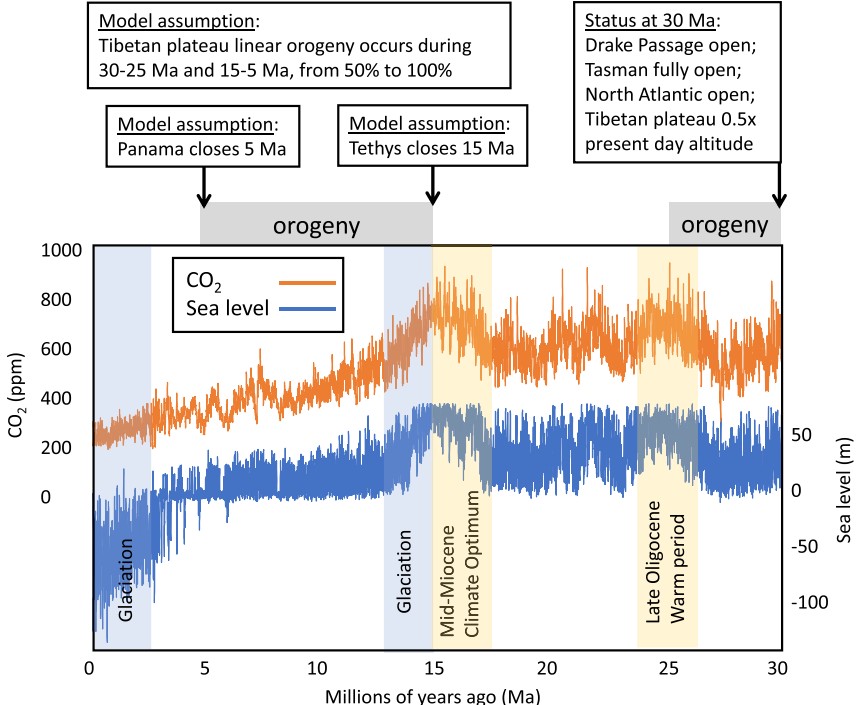

**Fig. 1 Major tectonic and climatic boundary conditions considered in the model for the past 30 Myr.** Changing boundary condition modelled are Himalayan-Tibetan Plateau (HTP) orogeny[15,28,34,41–44] (represented by a globally uniform scaling of orogeny), Tethys gateway closure[27], Panama gateway closure[61–64], orbital parameters[56], atmospheric $CO_2$[47,48] and the ice-sheet state derived from sea-level reconstructions[48]. Model assumptions include linear orogeny lifting between 30–25 Ma and 15–5 Ma from 50 to 100% of present day, final Tethys closure at 15 Ma and Panama closure at 5 Ma. The model assumes that Drake passage, Tasman and North Atlantic gateways were open at 30 Ma. See Supplementary Materials for discussion of data and references.

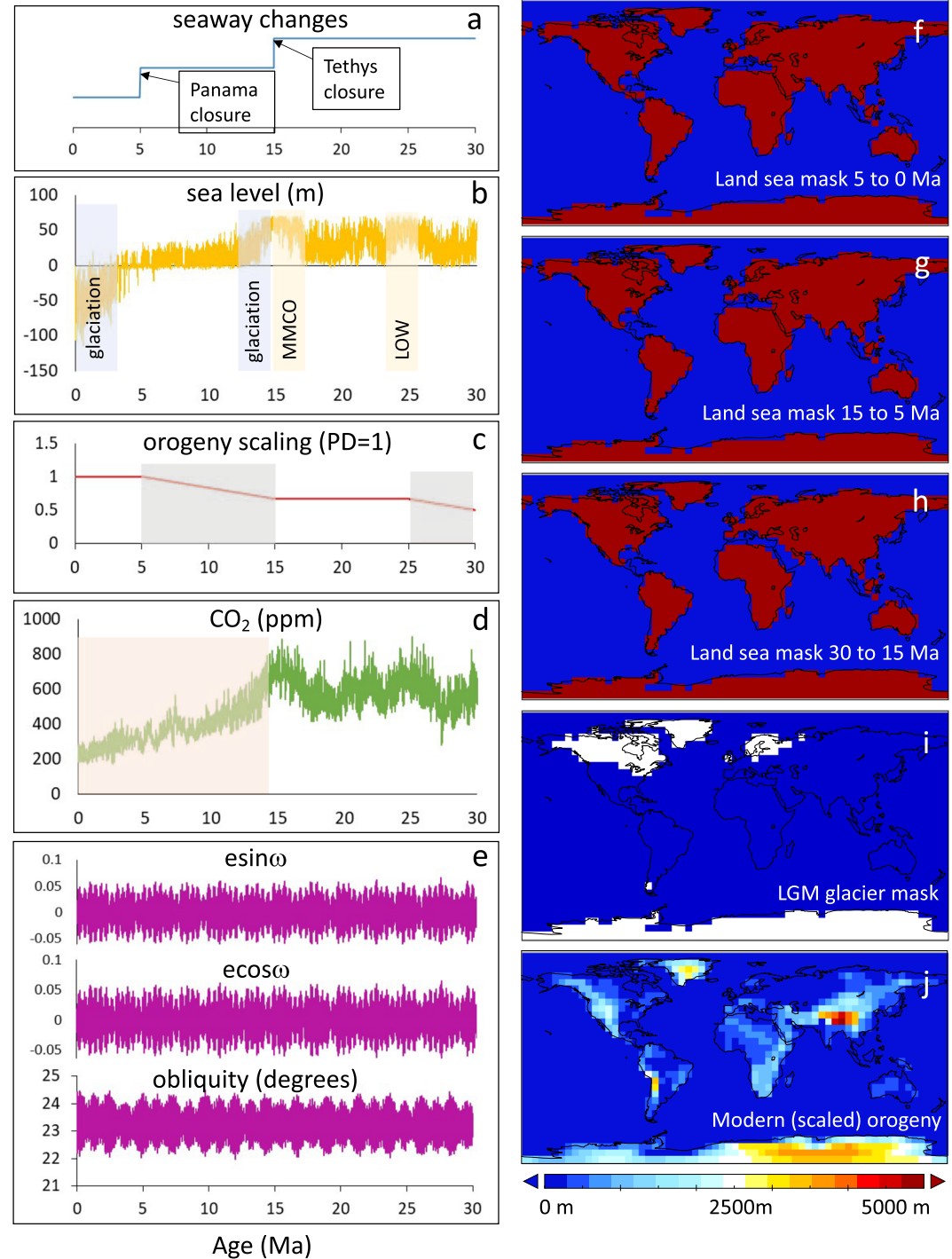

**Fig. 2 Boundary conditions used in this study.** Left panels are the time-series forcing, also used to derive priors for the total effect calulations (Supplementary Table 1), showing the assumed timing of (**a**) seaway land-sea mask changes, (**b**) sea-level[48] (MMCO Mid-Miocene Climate Optimum, LOW Late Oligocene Warming), (**c**) orogeny scaling (PD present day), (**d**) atmospheric $CO_2$[47,48] and (**e**) orbital forcing[56] (e eccentricity, ω longitude of the perihelion at the vernal equinox). Right-hand panels show the (**f**) 5 Ma to present, (**g**) 15–5 Ma, and (**h**) 30–15 Ma land-sea masks used in the study, (**i**) the Last Glacial Maximum (LGM) ice sheets[74] and (**j**) the modern orogeny.

Our global sensitivity analysis considers changes in tectonic, atmospheric $CO_2$, sea level (from which we infer the ice-sheet state) and orbital forcing since 30 Ma (Fig. 1; Table 1). The sensitivity analysis is derived from large ensembles of emulations, drawing inputs randomly from prior distributions that are derived from the forcing time series (Fig. 2 and Supplementary Table 1). We calculate the "total effect" of each input, which is the expectation of the variance that would remain if all other inputs

were known, and thus quantifies the sensitivity of the system to that input in isolation.

For SEAM rainfall, orogeny explains 93% of the variance over the last 30 Myr. Orogeny particularly dominated the rainfall variability during the Oligocene and Miocene[1,15] but eccentricity, obliquity, $CO_2$, sea level (ice-sheet state) and precession each contribute comparably to rainfall variability during the Plio-Pleistocene (when the model assumes attainment of present-day

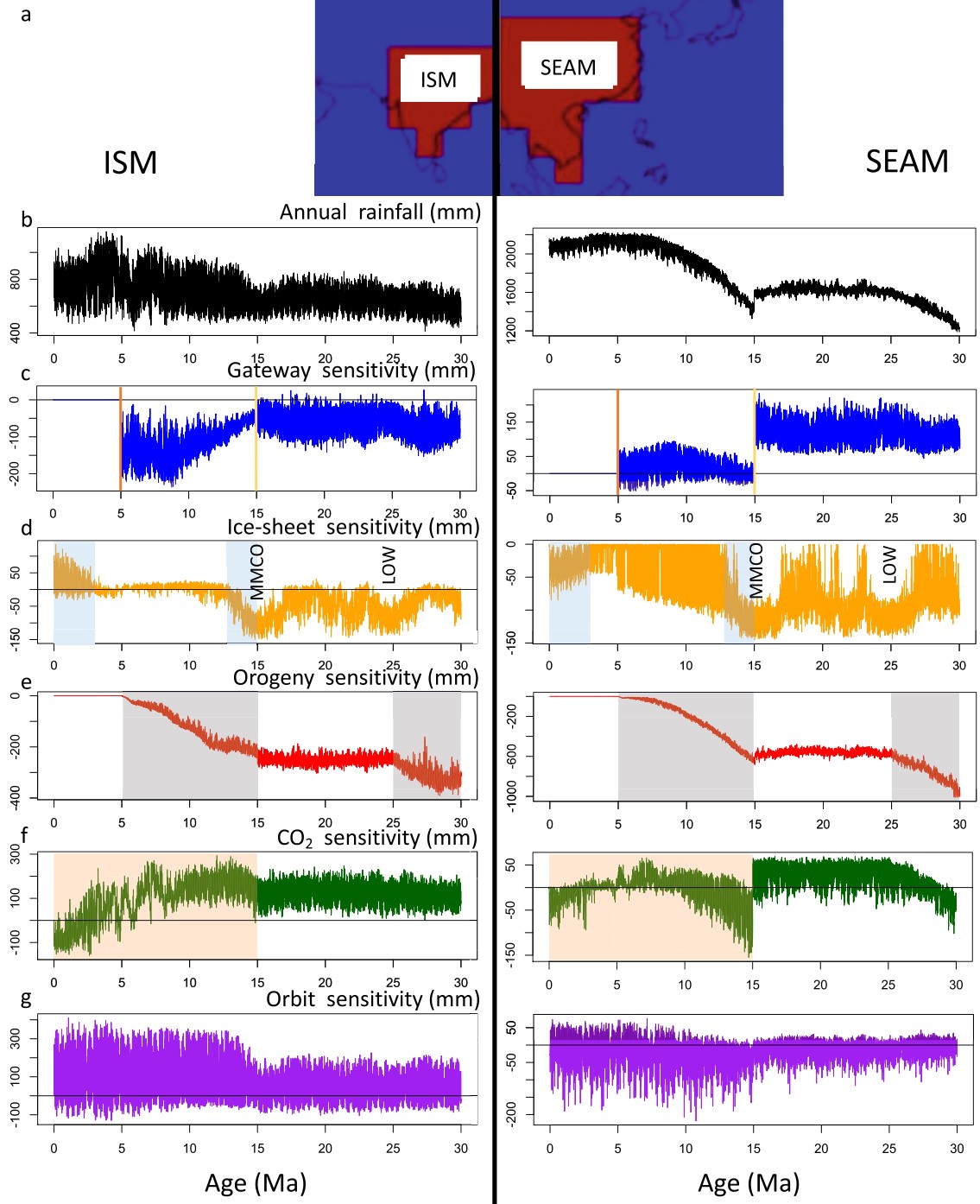

**Fig. 3 Monsoon rainfall and sensitivities since 30 Ma, showing the difference between the baseline rainfall and rainfall assuming the input is fixed at its preindustrial value. a** Modelled areas for Indian Summer Monsoon (ISM, left panel) and South East Asian Monsoon (SEAM, right panel). **b** Emulated rainfall. **c** Gateway sensitivity. Vertical lines are closing of Tethys gateway (yellow) and closing of Panama gateway (orange) Vertical labels are the Mid-Miocene Climate Optimum (MMCO) and Late Oligocene Warming (LOW). **d** Sea-level/ice volume sensitivity. Light blue shading are times of glaciation. **e** Orography sensitivity. Grey shaded areas are periods of tectonic uplift. **f** $CO_2$ sensitivity. Light red shading shows period of $CO_2$ drawdown. **g** Orbital sensitivity. Note the different y-axis scales used for the different forcings and for the two monsoon systems.

orography) (Table 1). Our Pleistocene results are in concert with terrestrial and marine records[5,8,16,17] suggesting multiple drivers (precession, obliquity, ice sheet and $CO_2$) impacting SEAM rainfall variability.

For ISM variability, precession is the dominant driver over the last 30 Myr, except during the Miocene, when orogeny and $CO_2$ are of comparable importance to (and amplify) precession, reflecting significant changes in these boundary conditions during

this period (Table 1). The dominance of precession on Pleistocene ISM variability has been demonstrated in cave[18] and marine[7] records. The greatest rainfall variance is apparent in the Pliocene, largely driven by the global $CO_2$ decrease (Table 1). Considering the entire 30 Myr interval, all of the modelled boundary conditions except obliquity played a significant role, with orogeny (39%), precession (25%) and $CO_2$ (21%) exerting the strongest controls on ISM rainfall.

**Table 1 Total effects on annual rainfall for each epoch (upper) ISM and (lower) SEAM.**

| | Holo-cene | Pleisto-cene | Pliocene | Miocene | Late Miocene 15.97–5.333 Ma | Early Miocene 23.03–15.97 Ma | Oligo-cene | 30 Ma to pre-industrial | 30 Ma to pre-industrial % of total |
|---|---|---|---|---|---|---|---|---|---|
| **ISM** | | | | | | | | | |
| CO$_2$ | 13 ± 2 | *44 ± 6* | **63 ± 11** | *55 ± 13* | **73 ± 4** | *14 ± 1* | *16 ± 1* | **71 ± 5** | 21% |
| Sea level | 9 ± 1 | 16 ± 1 | 14 ± 1 | *31 ± 3* | 28 ± 3 | *29 ± 1* | *24 ± 1* | 35 ± 3 | 5% |
| Precession | **127 ± 4** | **106 ± 5** | **123 ± 4** | 91 ± 5 | 91 ± 8 | 59 ± 3 | 55 ± 5 | 77 ± 5 | 25% |
| Obliquity | 11 ± 2 | 9 ± 1 | 12 ± 1 | 20 ± 1 | 18 ± 1 | *17 ± 1* | 13 ± 1 | 17 ± 1 | 1% |
| Eccentricity | 43 ± 1 | 36 ± 1 | 44 ± 0.4 | 32 ± 2 | 33 ± 3 | 25 ± 1 | 25 ± 1 | 30 ± 2 | 4% |
| Orogeny | 0 | 0 | 2 ± 0.1 | **65 ± 4** | **66 ± 4** | 0 | 20 ± 1 | **97 ± 3** | 39% |
| Gateways | – | – | – | – | – | – | – | 35 ± 1 | 5% |
| Total | 135 | 121 | 146 | 134 | 142 | 74 | 71 | 155 | 100% |
| **SEAM** | | | | | | | | | |
| CO$_2$ | 2 ± 0.1 | 20 ± 2 | *11 ± 0.4* | 23 ± 1 | 24 ± 1 | 6 ± 1 | 11 ± 0.3 | 31 ± 1 | 1% |
| Sea level | 9 ± 0.3 | 16 ± 0.2 | 13 ± 1 | 31 ± 1 | 33 ± 1 | **24 ± 1** | 24 ± 0.2 | 35 ± 1 | 1% |
| Precession | **39 ± 3** | **36 ± 3** | **38 ± 2** | 31 ± 1 | 30 ± 1 | **17 ± 1** | 15 ± 0.4 | 28 ± 1 | 1% |
| Obliquity | 14 ± 1 | 13 ± 1 | 10 ± 0.4 | 10 ± 1 | 11 ± 1 | 9 ± 1 | 8 ± 1 | 10 ± 1 | <1% |
| Eccentricity | 19 ± 4 | 18 ± 2 | 19 ± 1 | 20 ± 1 | 19 ± 1 | 14 ± 0.3 | 12 ± 1 | 17 ± 1 | <1% |
| Orogeny | 0 | 0 | 5 ± 1 | **184 ± 1** | **181 ± 1** | 0 | **109 ± 2** | **298 ± 1** | 93% |
| Gateways | – | – | – | – | – | – | – | 55 ± 1 | 3% |
| Total | 46 | 49 | 47 | 192 | 189 | 34 | 114 | 309 | 100% |

Total effects quantify the variance in precipitation that is attributable to the respective driver (methods). Total effects (aside from percentages) are provided as their square roots to aid comparison by converting this variance measure to units of mm/day. Mean and 1-sigma uncertainties are provided for each by repeating each calculation ten times with different (stochastically derived) emulators. Significant drivers in each epoch are highlighted; bold face text highlights drivers that contribute at least 20% variance to that epoch while italicised text highlights drivers that contribute 5–20% to the variance in that epoch. The gateways comprise the effects of Tethys and Panama closure. Over 30 Myr (RH column), SEAM rainfall has been dominated by orogeny, whereas ISM rainfall has been driven mostly by a combination of orogeny, precession and $CO_2$, with significant effects from sea level (ice volume) and ocean gateways. Note that while the Total Effects are dominantly a function of the prior of the parameter under question, they also depend upon the priors of the other variables through parameter interactions (methods). For this reason, the total effects in the Miocene cannot be derived from the values in the late and early Miocene, and may take values outside of their range.

To supplement the emulated sensitivity analysis we also performed six snapshot simulations with PLASIM-GENIE in order to better understand the underlying model dynamics. These simulations represent the preindustrial state and five sensitivities, having orography reduced by 50%, $CO_2$ doubled to 560 ppm, Last Glacial Maximum (LGM) ice sheets added, Panama gateway opened and precession in the opposing phase. The simulation outputs are provided in Supplementary Figs. 2–7, and used to inform the discussion that follows.

Our finding that orogeny dominates emulated SEAM rainfall, while multiple factors impact ISM rainfall is new but not surprising, because the ISM and SEAM are known to be distinct physical systems[19,20], and relatively higher sensitivity of SEAM to orogeny compared to ISM has been shown in proxy reconstructions[21]. This differing behaviour is supported by the single-forcing PLASIM-GENIE simulations (Supplementary Figs. 2–7). A 50% reduction in orogeny compared to pre-industrial drives a substantial weakening of the June-July-August (JJA) Hadley circulation at the longitude of the SEAM 107°E (Supplementary Fig. 4) and Walker circulation (Supplementary Fig. 7), and is associated with a reduction in JJA and annual rainfall (Supplementary Figs. 2 and 17) over the entire SEAM region. However, although the other four drivers affect the spatial distribution of SEAM rainfall, these changes are not associated with a substantial reorganisation of the Hadley circulation, and rainfall change averaged over the SEAM area has been dominated by orogeny. In contrast, significant changes in JJA Hadley circulation at 79°E (the approximate longitude of the ISM) are driven by each of the five forcings considered. Reduced orogeny, LGM ice sheets and opened Panama gateway all drive a weakened JJA Hadley circulation at 79°E and reduced ISM rainfall. LGM ice sheet[22] and open Panama gateway[23,24] are associated with a southerly shift of the ITCZ. In the case of the opened Panama gateway, the shift in the ITCZ is driven by Atlantic-Pacific salinity redistribution[23,24], collapse of the AMOC[25] (Supplementary Fig. 5) and redistribution of interhemispheric ocean heat content (Supplementary Fig. 6). Conversely, doubled $CO_2$ drives continental warming[26], which strengthens the Hadley circulation at 79°E and drives increased ISM rainfall. Similarly, a phase reversal in precession drives warming in the northern summer, a northerly shift of the JJA ITCZ, strengthened Hadley circulation and increased ISM rainfall.

**Monsoon rainfall changes since 30 Ma.** Monsoon rainfall is influenced both by direct effects such as physical changes to atmospheric flow driven by orogeny, $CO_2$ and Earth's orbital configuration and by indirect teleconnections, via atmospheric and oceanic circulation. These drivers influence the atmospheric circulation by both the meridional shifts in the ITCZ via Hadley Cells (affecting rainfall) as discussed in the previous section and zonal shifts of Walker Cells (affecting wind direction particularly in SE Asia[2]) or any other factors impacting them, for example, ocean circulation changes such as AMOC strength and Tethys Indian Saline Water (TISW)[27]. We discuss the impact of monsoon drivers, and their influences, on rainfall changes (Fig. 3b) and, using sensitivity analysis, isolate the contribution of each driver in turn (Fig. 3c–g) over the 30 Myr study interval by subtracting the results of a repeat emulation, fixing the parameter at preindustrial conditions.

For the period 30 Ma to 15 Ma, Fig. 3b shows ISM rainfall increased (~100 mm) due to the elevated orography, with lesser (temporary) effects due to reduced ice volume during the Late Oligocene Warming (LOW) and Mid-Miocene Climate Optimum (MMCO) (see Supplementary Fig. 1), and continuous oscillations due to orbital forcing. For the same period, SEAM rainfall increased significantly, mainly driven by increasing orography during 30–25 Ma, and with lesser effects of climate variability during LOW and MMCO. Orbital forcing had a much smaller effect on SEAM than ISM rainfall throughout the entire 30 Ma interval. The growth of the high altitude central and southern HTP (in both area and elevation) is likely to have occurred prior to 15 Ma[28] and so uplift tectonics impacts significantly on the SEAM during this period.

The period after 15 Ma is more complex with renewed orogeny, gateway closures, and ice-sheet (sea level) changes each having effects on both monsoon systems. Final Tethys closure at 15 Ma caused rapid, pronounced rainfall reduction in SEAM, with negligible effect on ISM rainfall (Fig. 3b, c). The step change in SEAM rainfall at 15 Ma, coincident with Tethys closure, seen in the $CO_2$ sensitivity (Fig. 3f) suggests an interaction between these two forcings i.e. when $CO_2$ is reduced to 280 ppm, emulated rainfall decreases when Tethys is closed, but increases when Tethys is open. We performed additional sensitivity simulations

to explore JJA rainfall for some of these interactions, using the same values as the single-parameter sensitivities, but combining them into eight combinations of Tethys open/closed, $CO_2$ 560 ppm/280 ppm and global orogeny scaling 1.0/0.5 (Supplementary Fig. 8). Tethys closure drives a simulated reduction in SEAM rainfall in all four scenarios, albeit with a substantial difference in magnitude and distribution, illustrating strong interaction. At 50% orogeny, reducing $CO_2$ from 560 ppm to 280 ppm reduces simulated annual rainfall from 1320 mm to 1300 mm when Tethys is closed. In contrast, when Tethys is open, reducing $CO_2$ from 560 ppm to 280 ppm increases annual rainfall from 1216 mm to 1242 mm. It is worth noting that these interactions are even more complex for the ISM, as Tethys closure drives rainfall change of uncertain sign, depending on the background state (Supplementary Fig. 8).

After Tethys closure, increasing orogeny (Fig. 3e) during the period 15 Ma to 5 Ma caused an increase in monsoon rainfall which has been inferred to be due to lower summer Asiatic atmospheric pressures[1] and strengthened Hadley-Walker circulation over East Asia[2], consistent with PLASIM-GENIE simulations (Supplementary Figs. 4 and 7). Furthermore, $CO_2$ drawdown began after 15 Ma and continued more-or-less until the Industrial Revolution (Fig. 1), driven by HTP weathering, and led to global cooling[29]. These three effects—atmospheric disturbance caused by orogeny, $CO_2$ drawdown, and increasing ice volume—produced competing effects in terms of monsoon rainfall. Both SEAM and ISM rainfall show increasing sensitivity to orbital forcing from about 15 Ma, peaking by around 12 Ma, associated with times of increasing Antarctic glaciation and orogeny forcing (Fig. 2).

During the period 14 Ma to 10 Ma, SEAM rainfall shows orbital variability synchronous with increases in ice volume and obliquity minima (Supplementary Fig. 10), in agreement with proxy data[3,30]. This strong obliquity signal on SEAM rainfall is likely due to the indirect effect of high-latitude teleconnection with northern hemisphere ice sheets, as the total effect analysis reveals a weaker relationship with obliquity than sea level (Table 1). SEAM rainfall peaked at c. 7 Ma, as orogeny approached its present-day elevation.

Panama closure at 5 Ma had minimal effect on SEAM rainfall but caused a dramatic increase of ~100 mm/year on ISM rainfall (Fig. 3c). We infer this ISM rainfall increase to be driven by indirect teleconnections arising from constriction of the Panama gateway, which led to increased AMOC[31] (Supplementary Fig. 5), redistribution of interhemispheric ocean heat content (Supplementary Fig. 6) and consequent northward movement of the ITCZ[32,33]. Falling $CO_2$ after 3 Ma drove a reduction in ISM rainfall (Fig. 3f), despite a competing increase driven by Pleistocene glaciation. The ice-sheet sensitivity simulation reveals a weakened JJA Hadley circulation (Supplementary Fig. 4) that drives reduced JJA precipitation (Supplementary Fig. 2), but this is offset in the annual average by an increase in DJF precipitation due to strengthened winds blowing onto the subcontinent from the Bay of Bengal. The relative contributions of these seasonal changes are not robust under parametric uncertainty (Supplementary Fig. 13) and should be treated with caution.

It is instructive to compare these long-term changes with a review of proxy reconstruction evidence[34] that concluded that the SEAM intensified between 25 and 20 Ma, putatively in response to northern HTP uplift. Further intensification has been associated with northeastern and eastern HTP uplift from 15 to 10 Ma[35,36], possibly masked by the competing effects of $CO_2$ drawdown, cooling and increasing glaciation[34]. Our emulations are broadly consistent with these inferences, although orographic forcing dominates in our results, and clear net strengthening of SEAM rainfall is emulated in response. Proxy evidence further

indicates transient increases in SEAM intensity at around 16–14 Ma and 4.2–2.8 Ma[34]. Similar events are seen in the ISM emulator at around these times (14–13 Ma and 4–3 Ma), but are absent in the SEAM emulator, which is dominated by orography in this period. During the Pleistocene, proxies indicate SEAM weakening, but with increased orbital variability[34]. This weakening is apparent (though modest) in the SEAM emulation, driven by a combination of ice-sheet and $CO_2$ forcing. Although increased variability is evident in our emulated records, this increase occurs earlier, over the period 15–13 Ma, possibly associated with the final HTP uplift, unrelated to Pleistocene boundary conditions. Orbital timescale monsoon variability has been evidenced in proxy records throughout the last 30 Ma and earlier[37–39], suggestive of both low-latitude precessional forcing and high-latitude obliquity forcing via ice sheets, and consistent with our emulators. A direct comparison of orbital variability between our regional-scale emulators and individual, locally-influenced proxy records would be extremely challenging[40], but possible in principle if a sufficient density of proxy reconstruction records could be assembled.

**Summary**. In summary, we conclude that the responses of the ISM and the SEAM to evolving forcing conditions are complex, driven by composite interactions between forcings. This complexity is especially evident in the ISM rainfall evolution, which is driven by competing climatic ice-sheet and $CO_2$ effects during the Miocene and Pliocene. These results need to be validated in the proxy-based ISM precipitation records from the Miocene through to the Pliocene at orbital resolution. Multiple records capturing the spatial and temporal variability in response to these climatic factors will strengthen our understanding of the precipitation response. We recommend that the two Asian Monsoon subsystems, SEAM and ISM, need to be considered as entirely distinct systems, and furthermore, the role of boundary conditions should be considered collectively in future studies. The ISM is especially sensitive to $CO_2$ change, and our results suggest that the scale of future changes in monsoon rainfall strength can be expected to be comparable to the scale of historical variations since at least the start of the Holocene, and probably comparable to that triggered by all forcing factors over the last 30 million years

## Methods

**Boundary conditions**. The boundary condition input data are summarised in Fig. 2.

The timing and extent of Himalayan-Tibetan Plateau (HTP) orogeny have been subject to sometimes conflicting assessments[15,28,34,41–46]. For modelling purposes, we have assumed that HTP orogeny occurred at a constant rate between 30–25 Ma and 15–5 Ma, doubling in height between 30 Ma and the present day. We also assumed that Himalayan-Tibetan orogeny was uniform within the entire HTP region, whereas the HTP expanded in area, as well as growing in height, at different times. These simplifying, or even simplistic, assumptions are justified on the basis that (a) the exact history of Tibetan orogeny remains uncertain, and (b) the objective in the present work is to evaluate the relative importance of all the potential factors, including orogeny, which could affect monsoon strength.

PLASIM-GENIE is a low-resolution model (~5 degrees), and as such the HTP is represented by only 5 × 2 grid cells (Fig. 2). A uniform uplift assumption is therefore most appropriate. We chose to apply this as a simple global scaling of present-day orography. As appropriate for a low-resolution model, we emulate rainfall changes on large spatial scales, averaged over the entire ISM and SEAM regions (Fig. 3a). More detailed spatial diagnostics, and assumptions of spatially variable orogeny growth, would require significantly higher model resolution, beyond what is tractable with current computing power, even with the benefit of emulation.

Ice-core $CO_2$ data are used when available[47], but prior to 800 ka $CO_2$ data[48] reconstructed from a zonally averaged energy balance climate model bidirectionally coupled to a one-dimensional ice sheet model were used; these capture ice–albedo and surface–height–temperature feedbacks. These data vary from others[49] that use a continuous alkenone-based $CO_2$ record from a single marine site, but which show a problematic decoupling of global climate from $CO_2$. The reconstructed $CO_2$ data[48] are broadly consistent with average global temperature trends, whereas the

alkenone-based data[49] are inconsistent with the Mid-Miocene Climate Optimum (MMCO)[50–52]. A possible partial explanation of this apparent anomaly has been proposed[53] suggesting caution in the interpretation of alkenone-based records at low $CO_2$ levels. A $CO_2$ peak during the MMCO has been ascribed to outgassing arising from Columbia River flood basalt, which dates between 16.7 and 15.9 Ma[54].

Sea level changes based on calculated global ice volumes were used[48]. Major increases in glaciation occurred, coincident with large falls in eustatic sea level, at ~34 Ma (onset of Antarctic glaciation coincident with opening of the Drake passage and formation of the Antarctic Circumpolar Current (ACC)), ~14 Ma (coincident with Greenland and West Antarctic glaciation), and ~3 Ma (the onset of Pleistocene glaciation)[55].

Orbital forcing data were taken from astronomical modelling[56]. Changes in vegetation types affect climate due to albedo and evapotranspiration feedbacks, but quantitative data over 30 Ma are not available. Although the self-shading parameter, which controls the maximum simulated vegetation density, was varied in the simulation ensemble, for emulation purposes vegetation was limited by self-shading to the tuned value[57] of 11.5 $m^2kgC^{-1}$.

Modelling used three gateway states: Panama and Tethys gateways both open (pre-15 Ma), Panama gateway only open (15–5 Ma), and modern world (post-5 Ma). Before 30 Ma, the Tarim basin had become dry[58], the Drake passage was open[23], Antarctic Circumpolar Circulation (ACC) was established[59], and North Atlantic Deep Water (NADW) was established (perhaps aided by Greenland-Scotland ridge subsidence[60]). Land-sea masks for the three modelled gateway states are illustrated in Fig. 2. The depths of the opened gateways are both assumed to be ~1000 m.

Final Tethys closure, modelled at 15 Ma[27], has been proposed as a driver of AMOC by reversing flow through the Gibraltar Strait. Tethys closure also led to global cooling by stopping the formation and circulation of Tethyan Indian Saline Water (TISW), which previously transported heat from the northern Indian Ocean to the Southern Ocean[27].

We assume Panama closure at 5 Ma, although evidence is inconclusive[61–64]. Unmodelled gateway changes include the Fram Strait opening to deep water at 18–14 Ma[65], Iceland formation at ~17 Ma[60], temporary closure of the Straits of Gibraltar 6–5.3 Ma[66], and Bering Strait opening (shallow) at ~5 Ma[67]. Constraint of Indonesian Through Flow (ITF or the Eastern Tethys) between Indian and Pacific Oceans began ~25 Ma, and this may have caused long-term El Niño leading ultimately to the Pliocene Warm Period, but the Makassar Strait remained deep enough that ITF was not cut off even during Pleistocene glaciation[68–71], so ITF constraint is not modelled here.

Supplementary Fig. 1 summarises the calculation methodology.

PLASIM-GENIE is a coupling of the intermediate-complexity spectral atmosphere model PLASIM to the Grid-Enabled Integrated Earth system model GENIE[4]. We applied the model at a spectral T21 atmospheric resolution (5.625 degrees) with 10 vertical layers, and a matching ocean grid with 16 logarithmically spaced depth levels, using the optimised parameter set[72]. We generate quasi-equilibrium 2000-year simulations in geared ocean-atmosphere mode[73], in which PLASIM-GENIE alternates between conventional coupling (for 1 year) and a fixed-atmosphere mode (for 9 years).

**Ensemble design.** Two 200-member ensembles were performed, each using a different maximin Latin hypercube design on eight varied parameters, being the ocean gateway configuration (present day, Panama open, both Tethys and Panama open), ice sheet state (present day, Last Glacial Maximum[74], only East Antarctic Ice Sheet, no ice sheets), the vegetation self-shading parameter[57] (from 0.5 to 20 $m^2kgC^{-1}$), global orography scaling (from 0 to 1 in the first ensemble, from 0.5 to 1 in the second ensemble), $CO_2$ concentration (varied with logarithmic spacing from 160 to 1500 ppm) and the three orbital parameters. The sampling strategy for the orbital variables (eccentricity, the longitude of the perihelion at the vernal equinox and obliquity) followed Araya-Melo[75], uniformly sampling in the range −0.07 to 0.07 (first ensemble) and −0.05 to 0.05 (second ensemble) and in the range 21° to 26° (first ensemble) 22° to 25° (second ensemble).

Emulators were constructed as Gaussian processes (GP)[76], using the *DiceKriging* R package[77]. GPs are highly flexible non-parametric regression models, widely used in the climate modelling community, which have greater modelling power than linear models. Best-fit emulators were determined under leave-one-out cross-validation, varying the covariance function, mean function, nugget value option, number of iterations, and convergence tolerance. $R^2$ was used as a measure of goodness-of-fit. Tests showed that using $esin\omega$ and $ecos\omega$ inputs yielded better performance than $e$ and $\omega$. Two outlying simulations were eliminated due to numerical instabilities in PLASIM-GENIE. The final emulators exhibit $R^2$ values of 80% for India and 96% for SE Asia under cross-validation.

Total effects[10] were calculated to assess quantitatively the relative importance of each parameter during each geological epoch. The total effect of an input parameter is the expectation of the output variance that would remain if all other parameters were known. For both India and SE Asia, two hundred ensembles, each with two hundred members, were performed for each parameter in each epoch. Each ensemble fixes the other parameters at some random value (drawn from their respective priors), while the tested parameter is varied across the ensemble (drawn from its prior). Different priors were derived for each epoch, considering the appropriate variability in the forcing data (Fig. 2 and Supplementary Table 1).

Normally distributed priors were used for all parameters except for *mvelp* and *oro* (uniform distributions) and *world* (which is selected as appropriate for each epoch). The mean variance of the 200 ensembles (for each parameter and epoch) is the total effect of that parameter in that epoch. The total effects for the whole period 30 Ma to preindustrial were also assessed, with the parameter *world* ascribed a value according to the relative duration of each world state over 30 Ma.

Time-series emulations were constructed in 1000-years steps over the last 30 Mya, using the forcing data time series in Fig. 2.

**PLASIM-GENIE validation.** The climate of PLASIM-GENIE has been validated against modern observations[4,73] and against model inter-comparisons of the mid-Holocene, the Last Glacial Maximum, the Last Interglacial transient and the mid-Pliocene warm period[72]. These analyses, which all used the 'optimised' parameter set to build the emulators, demonstrated that large-scale precipitation projections lie within the uncertainty envelope of high resolution IPCC-class models, which have themselves been validated against proxy data in the Mid-Holocene[78], the Last Glacial Maximum[78] and the Pliocene[79].

In this study, we have validated the simulated present-day monsoon system in our precise model setup in Supplementary Fig. 11, which compares global fields of precipitation (2005–2015) with NCEP reanalysis[80], and Supplementary Fig. 12 which compares the zonally averaged (70°E-90°E) distribution of modern simulated rainfall (2005–2015) through the seasonal cycle with both observations and high-resolution simulations[81]. Zonally averaged rainfall in PLASIM-GENIE peaks at 14 mm/day, slightly later (August) and at slightly lower latitudes (15°N) than observations, but generally falling within the range of high-resolution model behaviour. We note that the emulator calculates annual average rainfall over large spatial scales (Fig. 3) and therefore does not rely on the details of this spatiotemporal distribution.

**PLASIM-GENIE parametric uncertainty.** Model uncertainty is not captured by the emulator, because it was built from simulations with a single parameter set. In order to validate the robustness of the results under model uncertainty, we performed a series of perturbed parameter ensembles using a 69-member pre-calibrated parameter set[73]. These ensembles considered the forcings of orogeny (scaling by 50%), $CO_2$ (doubling to 560 ppm), precession (reversed phase) and LGM ice sheets and are summarised in Supplementary Fig. 13. Under parametric uncertainty, the three dominant drivers of uncertainty (Table 1), namely orogeny, $CO_2$ and precession, all drive ISM change that is significant with respect to the baseline preindustrial state and, despite significant uncertainty, all drive change of consistent sign. In all simulations, both doubled $CO_2$ and precession-reversal drive a strengthening of the monsoon while reduced orogeny drives a weakening. The optimised parameter set used to build the emulator lies close to the centre of all three ensembles. In contrast, the modest role of ice sheets (Table 1) is not robust under parametric uncertainty and should be treated with caution. Simulated SEAM rainfall is dominated by orogeny in all ensemble members. SEAM changes driven by $CO_2$ and precession are modest and (in contrast to the ISM) of uncertain sign, indicating that the phasing of orbitally-driven SEAM change should be treated with caution. We note that precession was found to be the dominant driver of Eocene monsoon variability simulated by PLASIM-GENIE in a fully altered paleogeography[82]. Reduced SEAM rainfall in response to LGM ice sheet is robust under parametric uncertainty.

**Boundary condition uncertainties.** Further uncertainties arise from the forcing boundary conditions. A similar simulation to our globally-scaled 30 Ma orogeny was performed in HadCM3[83], where global orogeny was reduced to 45% of modern relative to a world with a 1000 m Tibetan plateau but otherwise flat. Runoff from the Yangtze and Pearl rivers, which have catchment areas in the north of our modelled SEAM region, was found to reduce to approximately half of that under present-day orogeny, consistent with the rainfall reductions seen in the emulator (Fig. 3b) and the sensitivity simulation (Supplementary Figs. 2a and 17a). However, it is informative to examine the sensitivity with respect to the spatial distribution of the modelled orogeny. Supplementary Fig. 14 compares the distribution of northern summer rainfall in the baseline preindustrial simulation with simulations that flatten either the northern Tibetan Plateau or the southern Tibetan plateau, which includes the Himalayas. Flattening the northern plateau has a modest effects on both systems, while flattening the southern plateau and Himalayas substantially reduces SEAM rainfall. This is consistent with earlier work[84] which showed that the large-scale South Asian summer monsoon circulation is largely unaffected by removal of the Tibet plateau as long as the Himalayas are preserved. A closely related experiment[85], but at very high resolution (0.23°x0.31°), similarly concluded that the Himalayas are sufficient to sustain the Indian and Asian monsoon. High-resolution models are essential for detailed monsoon projections. For instance, 0.5° resolution has been found to be needed[86] to capture critical elements of the local monsoon moisture transport, a resolution which exceeds that of most state-of-the-art models. However, these comparisons suggest that large-scale monsoon features can be simulated at lower resolution, and furthermore that our emulators are unlikely to be sensitive to the details of the historic uplift. We note that simulations[87] in a 40 Ma paleogeography (pre-dating the assumed range of validity of our emulator), implied substantially altered atmospheric circulation patterns

from modern conditions, driving aridity over northern India and the HTP. While we capture elements of this in our scaled orography experiments (Supplementary Figs. 2a and 17a), the drying in PLASIM-GENIE is less pronounced and located further east. These differences may, at least in part, reflect the different orography assumptions.

A second important boundary condition uncertainty is the depth of Panama strait, which we have assumed to be ~1000 m. We performed a simulation with an 80 m deep straight, illustrated in Supplementary Fig. 15. In this simulation, freshening of the Atlantic does not penetrate to high northern latitudes and does not drive a weakening of the Atlantic meridional overturning circulation, nor of the Asian monsoon (in fact they both show increased strength, associated with increased surface Atlantic salinity at high latitudes). This illustrates, perhaps unsurprisingly, that the emulated step-change in ISM rainfall associated with Panama closure (Fig. 3) is a highly simplified representation of the transition.

In Supplementary Fig. 16, we consider a 3.5 $Wm^{-2}$ reduction in the solar constant under 30 Ma boundary conditions, an effect which was neglected in the emulators. Simulated annual SEAM rainfall is unchanged, ISM annual rainfall is reduced by 28 mm, and there are changes in the seasonal distributions of both systems. However, these changes are modest (implied $\sqrt{}$ total effect ~ 10 mm) compared to the other drivers (Table 1).

A further source of boundary condition uncertainty, distinct from the implementation choices, are the uncertain timings and magnitudes of the forcings. We present emulator sensitivities in Supplementary Figs. 18, 19 and 20, which test the effects of Panama closure timing (7–3 Ma), Tethys closure timing (17–13 Ma), atmospheric $CO_2$ (reduced to 560 ppm at the MMCO, see Supplementary Fig. 18) and changed orogeny timing (see Supplementary Fig. 18). The low-start linear orogeny approximates the growth of the Himalayas[88], included because of the sensitivity of the simulated Asian monsoon circulation to the Himalayan orogeny[69]. These emulations do not show any unexpected behaviour, but serve to demonstrate the range of sensitivity to uncertain timing assumptions.

## Data availability
All relevant data are available from the authors. Source data are provided with this paper.

## Code availability
The PLASIM-GENIE model is open access software, implemented in 'cgenie.muffin', which can be found at https://github.com/derpycode/cgenie.muffin. The emulations were performed in R. The R code and input data are provided in the appendix appendix.tar.gz.

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

## Acknowledgements

P.H. and N.R.E. acknowledge NERC funding (grant number NE/P015093/1). P.A. acknowledges NERC and UKIODP funding (NE/M021181/1). We thank Clare Warren and Robert Spicer for careful reading of the manuscript.

## Author contributions

P.A. and P.B.H. conceived and designed the experiment. P.B.H. performed the PLASIM-GENIE simulations. J.R.T. developed the emulators and performed all of the emulation analysis with guidance from P.B.H. and N.R.E.. C.A.P. developed the boundary condition data and early versions of the emulation approach. Scientific expertise was provided by N.B.W.H. (Himalayan and Tibetan tectonics and Asian monsoon), P.A. (Asian Monsoon dynamics), P.B.H. (Earth system modelling and emulation) and N.R.E. (Earth system dynamics). J.R.T. wrote the manuscript with contributions from all authors.

## Competing interests

The authors declare no competing interests.
