## [Peer Review File · Nature Communications]

Reviewers' Comments:

Reviewer #1:

Remarks to the Author:

Review of: Tectonic and climatic drivers of Asian monsoon evolution

By James R. Thomson, et al.

Overview:

Thomson, et al. attempt to reconstruct the dominant forcing's and driving mechanisms behind the evolution of the Asian summer monsoon system, sub-divided into both the Indian and East Asian monsoon. As noted, it is difficult to ascertain the relative importance of different processes over geologic time which this work tries to address in a consistent manner. This is an interesting piece of work presented and confirms other recent studies on the various importance of certain boundary conditions on the response of the Asian monsoon system. I in particular like the attempt to investigate orbital variability as this is an arena that has seen far less attention over deep-time. I think this work has merit and eventually could be published; however, I have some concerns that would need to be addressed first.

Major comments:

The biggest concern is that this work is driven by a modern geography only. With paleogeography changing considerably over the last 30 million years, including (globally and regionally) oceanic gateways (e.g. Drake's passage, Turgai strait, Panama isthmus) opening and closing, the closure and disappearance of the Paratethys, change in solar energy, ice sheets forming in the Southern and then Northern Hemisphere. While Thomson, et al. attempt to investigate some of these issues, they still use a modern geography and not those past time periods. Continents, mountain belts, deserts, ocean basins were all very different to that of today and would have a dissimilar impact on the Asian monsoon. This needs to be discussed more as these issues will have an impact on the thermodynamics of the monsoon system. Presently, this work mainly shows the impact of changing certain boundary conditions on the modern Asian monsoon system only.

Model skill. How skillful is PLASIM-GENIE at reproducing the modern ISM and EASM statistics? It is essential to know so we can contextualize these results. An analysis of this would strengthen the conclusions of this ms.

Model resolution. Acosta and Huber (2020, 2017) (as well as others) have suggested that model resolution plays an important role in the ability to accurately simulate the characteristics of the Asian monsoon system. This needs further discussion and assessing the model skill wrt to the modern monsoon may help with this (if indeed the model does show skill).

I may have missed it but I cannot see whether change in solar energy was incorporated into the emulation and GCMs? There would have been a 3-5 W/m² change over this 30 Ma. period. What influence might this have had?

There is a lack of discussion on proxy-studies relating to orbital influences. I am left wondering whether the dominance of certain orbital signals is a result of the modern (although modified for sensitivity analysis) geography or whether these signals have been observed over the past 30 Ma. This would really add to the discussion and help contribute to the validity of the modelling results.

Model dependency. It is well known that different models can produce different results. While I would certainly not expect the authors to run the same set of simulations with different models, it would be nice to acknowledge that this. More so, it would also help comparing these results with other similar modelling work that is out there (e.g. Boos and Kuang, Acosta and Huber, Tardiff, et al., Farnsworth,

et al. 2019).

Proxy data. The real weakness of this study is a lack of Proxy data. I would like to see some attempt to evaluate the response suggested by this emulation and GCM approach. Presently it is hard to determine how much confidence we can have in these results when it is difficult to see whether the model and emulator are accurately able to reconstruct past time periods and change (which will be difficult given the modern, but modified land geography). There are a host of palaeoflora and biogeochemical evidence out there to do this. Again, addition and discussion of this would help justify these conclusions.

Tectonic changes. There seems to my knowledge be two competing tectonic considerations being presented with regards to Tibetan orogeny and evolution through the Cenozoic. Botsyun, et al. use a new and innovative approach to suggest a 'low' plateau rising while Spicer, et al. suggest two large mountain ranges with a low valley in between infilling. I am also not sure the reference cited does suggest that "modelled rainfall is insensitive to the details of the paleogeographic reconstruction", from looking at this reference they show the opposite and that CO₂ was not a big factor. Some discussion on the implications of resolution and the potential impacts of different HTP should be discussed. More so, Acosta and Huber also show that while a monsoon will always be present regardless of the addition of Tibet or not, the details do make a huge difference in the direction of moist flow in the Asian monsoon region. This brings me back to the resolution issue of the GENIE. Acosta and Huber (2017, 2020) show that you cannot get a realistic monsoon developing using current low-resolution GCMs let alone even coarser intermediate models such as GENIE. This needs to be discussed in some manner in this manuscript.

References:

- R.P. Acosta, M. Huber. (2020). Competing Topographic Mechanisms for the Summer Indo-Asian Monsoon *Geophysical Research Letters*, 47
- Acosta, R. P., & Huber, M. (2017). The neglected Indo-Gangetic Plains low-level jet and its importance for moisture transport and precipitation during the peak summer monsoon. *Geophysical Research Letters*, 44, 8601– 8610
- Boos, W., & Kuang, Z. (2010). Dominant control of the South Asian monsoon by orographic insulation versus plateau heating. *Nature*.
- S. Botsyun, P. Sepulchre, Y. Donnadieu, C. Risi, A. Licht, J. K. Caves Rügenstein. (2019). Revised paleoaltimetry data show low Tibetan Plateau elevation during the Eocene. *Science* 363.
- Delphine Tardif, Frédéric Fluteau, Yannick Donnadieu, Guillaume Le Hir, Jean-Baptiste Ladant, et al. 2020. The origin of Asian monsoons: a modelling perspective. *Climate of the Past*, 16 (3), pp.847-865

Reviewer #2:

Remarks to the Author:

This is a review of the manuscript „Tectonic and climatic drivers of Asian monsoon evolution“ by Thomson et al. The manuscript investigates how different impacts, ice sheets, tectonics, CO₂, orbital variability, have influenced changes in monsoon rainfall for the last 30 Myr using a model setup allowing to distinguish the impact of each separate factor. The main focus is on changes in the Indian Summer Monsoon (ISM) and the South East Asia Monsoon (SEAM). This study deals with a very important part of monsoon reconstructions as it focuses on changes in rainfall during the monsoon rather than changes in wind strength. Most proxies used for paleo-reconstructions are reconstructing changes in monsoon winds (e.g. Kroon et al., 1991; Gupta et al., 2015; Betzler et al., 2016), while changes in rainfall are difficult to reconstruct (e.g. Dettman et al., 2001; Huang et al., 2017). As such, this study would be a very good addition to available literature. I was a bit confused by the "data-modelling" approach mentioned in the abstract, because this made me think new proxy data would be presented too. However, this leads to a final bit I am missing in the discussion on how these results

support or counter evidence from proxy records. For several of the impacts you write that the literature is not conclusive, so could you now provide model evidence for a more likely scenario for Miocene changes in monsoon intensity?

I have to point out that I am not a modeler myself, so I cannot judge if the model setup is correct or not. What I do wonder about though is how some of the sensitivity experiments are determined. Are these based on variable results from existing proxy reconstructions or rather to set a range in boundary conditions? For example closure of Panama at 5 Ma, while open is defined as 1000 m water depth. This is a very broad definition. The Montes papers claim closure by the mid-Miocene, which probably relates to deep water cutoff. But most of the literature on the direct effects of "effective" Panama closure (e.g. Haug and Tiedemann, 1998; Osborne et al., 2014; modelling studies by Haywood, Lunt, Dolan) use the interruption of surface water masses in the late Pliocene from 4 Ma to the NHG. This brings in a range of possibilities that could have a large impact on your results. Closing of Panama has often been associated to provide the settings for Northern Hemisphere Glaciation as a result of the cutoff of mixed layer water coming from the Pacific into the Atlantic.

Similarly the final closure of the Tethys at the MCT, which some studies claim that TISW cessation may have triggered the MCT, but others say that the final closure was caused by the Antarctic glaciation through its sea level drop closing the final shallow bits of the Tethys, and thus that TISW already stopped much earlier. So the effect of a 1000 m deep gateway placed at the MCT or rather several million years earlier may make a large difference.

I suggest to have a look at the Raitzsch et al. (2020) paper and references in there dealing with atmospheric CO₂ concentrations from the MMCO crossing the MCT. The values in there are determined using B-isotopes in the shells of foraminifera. Based on these results a doubling of CO₂ in comparison with pre-industrial seems unlikely.

An interesting point is that changes in the Hadley circulation seem mainly have an influence on the ISM, but not on the SEAM area. The discussion on explaining this impact seems to be mainly focused on factors related to changes in the northern hemisphere, e.g. ITCZ movement due to AMOC collapse or LGM ice sheets. But how realistic is this for the Miocene situation? I do not think that AMOC collapse was a big issue then, and ice sheet changes mainly occurred on Antarctica, which would then suggest a northward shift of the Westerlies and the ITCZ/Hadley circulation (Groeneveld et al., 2017).

How was the choice for the height of the HTP for specific time intervals determined, especially the interval 15-5 Ma? Despite the uncertainties most studies do seem to agree that the HTP is there by 15 Ma (see also Ding et al., 2017). How does a different scenario for this time interval change the results?

Minor comments:

Figure 1: please put both curves on their own y-axis.

p.5 ref. 44: Kienast et al. Is cited here, but that seems wrong to me. This paper has nothing to do with the Pliocene closure of Panama.

Figure 2: why are the orogeny and gateway sensitivity records not continuing after 5 Ma?

To conclude, I think this manuscript would be a great addition for Nat. Comm. as it is dealing with a topic that currently receives a lot of attention, but I am missing some detail on the origin of the data used or rather on the different possibilities created by different datasets sometimes caused by missing literature. And an outlook or implications part at the end would allow the manuscript to be used for future proxy reconstructions.

Tectonic and climatic drivers of Asian monsoon evolution

James R. Thomson et al.

We are grateful for the detailed and constructive comments of both reviewers. We have now performed extensive additional analysis, as suggested by both, which has greatly strengthened the robustness of our conclusions. We have validated the simulated monsoon in both modern and paleoclimate states, quantified parametric uncertainty (model errors) through new simulation ensembles, tested simulated sensitivity to neglected and perturbed boundary conditions, and explored emulated uncertainty in response to the timings and magnitudes of boundary condition forcing. We regard a detailed comparison with proxies as outside of scope, as discussed below, but include a comparison with HadCM3 simulations and semi-quantitative proxy data (Farnsworth et al 2019).

Our responses are below, with reviewer comments in bold face, our responses in red type and revisions to the text in green. There are significant overlaps in the two reviews, so we have combined the responses, and most points are dealt with in the response to Reviewer #1.

Reviewer #1 (Remarks to the Author):

Thomson, et al. attempt to reconstruction the dominant forcing's and driving mechanisms behind the evolution of the Asian summer monsoon system, sub-divided into both the Indian and East Asian monsoon. As noted, it is difficult to ascertain the relative importance of different processes over geologic time which this work tries to address in a consistent manner. This is an interesting piece of work presented and confirm other recent studies on the various importance of certain boundary conditions on the response of the Asian monsoon system. I in particular like the attempt to investigate orbital variability as this is an arena that has seen far less attention over deep-time. I think this work has merit and eventually could be published; however, I have some concerns that would need to be addressed first.

Major comments:

The biggest concern is that this work is driven by a modern geography only. With paleogeography changing considering over the last 30 million years, including (globally and regionally) oceanic gateways (e.g. Drakes passage, Turgai strait, Panama isthmus) opening and closing, the closure and disappearance of the Paratethys, change in solar energy, ice sheets forming in the Southern and then Northern Hemisphere. While Thomson, et al. attempt to investigate some of these issues, they still use a modern geography and not those past time periods. Continents, mountain belts, deserts, ocean basins where all very different to that of today and would have a dissimilar impact on the Asian monsoon. This needs to be discussed more as these issues will have an impact on the thermodynamics of the monsoon system. Presently, this work mainly shows the impact of changing certain boundary conditions on the modern Asian monsoon system only.

Our approach, using intermediate complexity modelling and emulation, has enabled us to explore many drivers simultaneously, far more comprehensively than any previous study to our knowledge. We have included the most important drivers, including (from the reviewer's list) Panama closure, Tethys closure, ice-sheets in both hemispheres and evolving orogeny. Deserts are an emergent property of the vegetation model, and so these are also included. We

have added sensitivity analysis (see below) that demonstrates solar variability is negligible with respect to the drivers we have included. We have also added new sensitivities to the Tibetan plateau and Himalayan orogeny, and to the bathymetry of the Panama gateway. We did not consider either the Drake passage opening or the Turgai strait closing as these events are generally accepted to have occurred before 30Ma, which made them low priority, and would have been major undertakings to simulate and emulate.

We have now added a new section on these uncertainties in the Methods:

Boundary condition uncertainties

Further uncertainties arise from the forcing boundary conditions. A similar simulation to our globally-scaled 30Ma orogeny was performed in HadCM3⁷¹, where global orogeny was reduced to 45% of modern relative to a world with a 1000m Tibetan plateau but otherwise flat. Runoff from the Yangtze and Pearl rivers, which have catchment areas in the north of our modelled SEAM region, was found to reduce to approximately half of that under present-day orogeny, consistent with the rainfall reductions seen in the emulator (Fig 2b) and the sensitivity simulation (Fig S3a). However, it is informative to examine the sensitivity with respect to the spatial distribution of the modelled orogeny. Fig S15 compares the distribution of northern summer rainfall in the baseline preindustrial simulation with simulations that flatten either the Tibet Plateau or the Himalayas. Flattening the Tibet plateau has a modest effect on Asian monsoon rainfall, while flattening the Himalayas substantially reduces SEAM rainfall. This is consistent with earlier work⁷² which showed that the large-scale South Asian summer monsoon circulation is largely unaffected by removal of the Tibet plateau as long as the Himalayas are preserved. A closely related experiment⁷³, but at very high resolution ($0.23^{\circ} \times 0.31^{\circ}$), similarly concluded that the Himalayas are sufficient to sustain the Indian and Asian monsoon. High-resolution models are essential for detailed monsoon projections. For instance, 0.5° resolution has been found to be needed⁷⁴ to capture critical elements of the local monsoon moisture transport, a resolution which exceeds that of most state-of-the-art models. However, these comparisons suggest that large-scale monsoon features can be simulated at lower resolution, and furthermore that our emulators are unlikely to be sensitive to the details of the historic uplift.

A second important boundary condition uncertainty is the depth of Panama strait, which we have assumed to be ~1000m. We performed a simulation with an 80m deep straight, illustrated in Fig S16. In this simulation, freshening of the Atlantic does not penetrate to high northern latitudes and does not drive a weakening of the Atlantic meridional overturning circulation, nor of the Asian monsoon (in fact they both show increased strength, associated with increased surface Atlantic salinity at high latitudes). This illustrates, perhaps unsurprisingly, that the emulated step-change in ISM rainfall associated with Panama closure (Fig 2) is a highly simplified representation of the transition.

In Fig S17, we consider a 3.5Wm^{-2} reduction in the solar constant under 30Ma boundary conditions, an effect which was neglected in the emulators. Simulated annual SEAM rainfall is unchanged, ISM annual rainfall is reduced by 28mm, and there are changes in the seasonal distributions of both systems. However, these changes are modest (implied $\sqrt{\text{total effect}} \sim 10\text{mm}$) compared to the other drivers (Table 1).

A further source of boundary condition uncertainty, distinct from the implementation choices, are the uncertain timings and magnitudes of the forcings. We present emulator sensitivities in

Figs S18, S19 and S20, which test the effects of Panama closure timing (7 to 3Ma), Tethys closure timing (17 to 13Ma), atmospheric CO₂ (reduced to 560ppm at the MMCO, see Fig S18) and changed orogeny timing (see Fig S18). The low-start linear orogeny approximates the growth of the Himalayas⁷⁵, included because of the sensitivity of the simulated Asian monsoon circulation to the Himalayan orogeny. These emulations do not show any unexpected behaviour, but serve to demonstrate the range of sensitivity to uncertain timing assumptions.

Model skill. How skillful is PLASIM-GENIE at reproducing the modern ISM and EASM statistics? It is essential to know so we can contextualize these results. An analysis of this would strengthen the conclusions of this ms.

Thanks, we agree, this is very helpful. We have added this (and other validations) in a new section in the Methods:

PLASIM-GENIE validation

The climate of PLASIM-GENIE has been validated against modern observations^{4,62} and against model inter-comparisons of the mid-Holocene, the Last Glacial Maximum, the Last Interglacial transient and the mid-Pliocene warm period⁶¹. These analyses, which all used the ‘optimised’ parameter set used to build the emulators, demonstrated that large-scale precipitation projections lie within the uncertainty envelope of high resolution IPCC-class models, which have themselves been validated against proxy data in the Mid-Holocene⁶⁶, the Last Glacial Maximum⁶⁶ and the Pliocene⁶⁷.

In this study, we have validated the simulated present-day monsoon system in our precise model setup in Fig S12, which compares global fields of precipitation (2005-2015) with NCEP reanalysis⁶⁸, and Fig S13 which compares the zonally averaged (70°E-90°E) distribution of modern simulated rainfall (2005-2015) through the seasonal cycle with both observations and high-resolution simulations⁶⁹. Zonally-averaged rainfall in PLASIM-GENIE peaks at 14mm/day, slightly later (August) and at slightly lower latitudes (15°N) than observations, but generally falling within the range of high-resolution model behaviour. We note that the emulator calculates annual average rainfall over large spatial scales (Fig 1) and therefore does not rely on the details of this spatiotemporal distribution.

Model resolution. Acosta and Huber (2020, 2017) (as well as others) have suggested that model resolution plays an important role in the ability to accurately simulate the characteristics of the Asian monsoon system. This needs further discussion and assessing the model skill wrt to the modern monsoon may help with this (if indeed the model does show skill).

Thanks again, addressed in the new “Boundary condition uncertainties” section above.

I may have missed it but I cannot see whether change in solar energy was incorporated into the emulation and GCMs? There would have been a 3-5 W/m² change over this 30 Ma. period. What influence might this have had?

Thanks, good point, also addressed in “Boundary condition uncertainties” above.

There is a lack of discussion on proxy-studies relating to orbital influences. I am left wondering whether the dominance of certain orbital signals is a result of the modern (although modified for sensitivity analysis) geography or whether these signals have been observed over the past 30 Ma. This would really add to the discussion and help contribute to the validity of the modelling results.

We are not aware of any proxy data at sufficient resolution to demonstrate orbital variability in the early record. However, the dominance of orbital influences is not the result of geography in this model, and this is also now discussed within the new section on “PLASIM-GENIE parametric uncertainty” (see below) re the Eocene simulations of Keery et al (2018).

Model dependency. It is well known that different models can produce different results. While I would certainly not expect the authors to run the same set of simulations with different models, it would be nice to acknowledge that this. More so, it would also help comparing these results with other similar modelling work that is out there (e.g. Boos and Kuang, Acosta and Huber, Tardiff, et al., Farnsworth, et al. 2019).

Another excellent point, which we have now addressed through parametric uncertainty. We have performed several perturbed parameter ensembles to address uncertainties in the model responses to the dominant drivers. These are detailed in a new section, pasted below.

PLASIM-GENIE parametric uncertainty

Model uncertainty is not captured by the emulator, because it was built from simulations with a single parameter set. In order to validate the robustness of the results under model uncertainty, we performed a series of perturbed parameter ensembles using a 69-member pre-calibrated parameter set⁶². These ensembles considered the dominant forcings (Table 1) of orogeny (scaling by 50%), CO₂ (doubling to 560ppm) and precession (reversed phase) and are summarised in Fig S14. Under parametric uncertainty, all three forcings drive ISM change that is significant with respect to the baseline preindustrial state and, despite significant uncertainty, all drive change of consistent sign. In all simulations, both doubled CO₂ and precession-reversal drive a strengthening of the monsoon while reduced orogeny drives a weakening. The optimised parameter set used to build the emulator lies close to the centre of all three ensembles. In contrast, and consistent with the emulator, simulated SEAM rainfall is dominated by orogeny in all ensemble members. SEAM changes driven by CO₂ and precession are modest and (in contrast to the ISM) of uncertain sign, indicating that the phasing of orbitally-driven SEAM change should be treated with caution. We note that precession was also found to be the dominant driver of Eocene monsoon variability simulated by PLASIM-GENIE in a fully altered paleogeography⁷⁰, thus the details of the geography do not appear to be important.

The “Boundary condition uncertainties” section above includes comparisons with related modelling work as suggested (Boos and Kuang 2010, Lunt et al 2010, Acosta and Huber 2017, Acosta and Huber 2020, Keery et al 2018).

Proxy data. The real weakness of this study is a lack of Proxy data. I would like to see some attempt to evaluate the response suggested by this emulation and GCM approach. Presently it is hard to determine how much confidence we can have in these results when it is difficult to see whether the model and emulator are accurately able to

reconstruct past time periods and change (which will be difficult given the modern, but modified land geography). There are a host of palaeoflora and biogeochemical evidence out there to do this. Again, addition and discussion of this would help justify these conclusions.

The general conclusions of our emulations are consistent with the general inferences from proxy data, such as a strengthening through time, presumed in response to orogeny, and variability at orbital time scales that is driven, whether directly or indirectly (e.g. ice sheets and CO₂), by precession and obliquity (An Zhisheng et al 2015). However, the interpretation of proxy records is complex, given dating uncertainties which significant on orbital time scales, differences between local-scale responses which proxy records may reflect (Nilsson-Kerr et al 2019), interactions between forcing mechanisms, and the limited number of records that directly capture precipitation. These problems also apply to the long-term monsoon evolution and the wide range of proxies which have been used are often contradictory in both magnitude and timing (Lunt et al 2010). A full model-data consensus reconstruction would be a major and highly challenging undertaking, well beyond the scope of this paper. However, we have plotted a comparison below, of our SEAM rainfall emulations averaged over 1Ma time intervals (superposed orange circles in lower panel) against the simulations (lower) and semi-quantitative proxy data (upper) in Farnsworth et al (2019). The comparison shows remarkable similarity in precipitation trend for SEAM.

An Zhisheng et al (2015) Global monsoon dynamics and climate change, *Annu. Rev. Earth Planet. Sci.* 43, 29-77

K. Nilsson-Kerr, P. Anand, P. F. Sexton, M. J. Leng, S. Misra, S. C. Clemens, S. J. Hammond, Role of Asian summer monsoon subsystems in the inter-hemispheric progression of deglaciation. *Nat. Geosci.* **12**, 290 (2019).

Lunt, D.J., Flecker, R. and Clift, P.D. (2010) The impacts of Tibetan uplift on paleoclimate proxies, in Clift, P. D., Tada, R. & Zheng, H. (eds) *Monsoon Evolution and Tectonics–Climate Linkage in Asia*. Geological Society, London, Special Publications, 342, 279–291.

Tectonic changes. There seems to my knowledge be two competing tectonic considerations being presented with regards to Tibetan orogeny and evolution through the Cenozoic. Botsyun, et al. use a new and innovative approach to suggest a 'low' plateau rising while Spicer, et al. suggest two large mountain ranges with a low valley in

between infilling. I am also not sure the reference cited does suggest that “modelled rainfall is insensitive to the details of the paleogeographic reconstruction”, from looking at this reference they show the opposite and that CO₂ was not a big factor. Some discussion on the implications of resolution and the potential impacts of different HTP should be discussed. More so, Acosta and Huber also show that while a monsoon will always be present regardless of the addition of Tibet or not, the details do make a huge difference in the direction of moist flow in the Asian monsoon region. This brings me back to the resolution issue of the GENIE. Acosta and Huber (2017, 2020) show that you cannot get a realistic monsoon developing using current low-resolution GCMs let alone even coarser intermediate models such as GENIE. This needs to be discussed in some manner in this manuscript.

Thanks again, and also addressed in new section on “Boundary condition uncertainties”. We have removed the sentence “In addition, some recent simulation work has shown that modelled rainfall is fairly insensitive to the details of paleogeographic reconstruction.”

References:

R.P. Acosta, M. Huber. (2020). Competing Topographic Mechanisms for the Summer Indo-Asian Monsoon *Geophysical Research Letters*, 47

Acosta, R. P., & Huber, M. (2017). The neglected Indo-Gangetic Plains low-level jet and its importance for moisture transport and precipitation during the peak summer monsoon. *Geophysical Research Letters*, 44, 8601– 8610

Boos, W., & Kuang, Z. (2010). Dominant control of the South Asian monsoon by orographic insulation versus plateau heating. *Nature*.

S. Botsyun, P. Sepulchre, Y. Donnadieu, C. Risi, A. Licht, J. K. Caves Rügenstein. (2019). Revised paleoaltimetry data show low Tibetan Plateau elevation during the Eocene. *Science* 363.

Delphine Tardif, Frédéric Fluteau, Yannick Donnadieu, Guillaume Le Hir, Jean-Baptiste Ladant, et al. 2020. The origin of Asian monsoons: a modelling perspective. *Climate of the Past*, 16 (3), pp.847-865

Reviewer #2 (Remarks to the Author):

This is a review of the manuscript „Tectonic and climatic drivers of Asian monsoon evolution“ by Thomson et al. The manuscript investigates how different impacts, ice sheets, tectonics, CO₂, orbital variability, have influenced changes in monsoon rainfall for the last 30 Myr using a model setup allowing to distinguish the impact of each separate factor. The main focus is on changes in the Indian Summer Monsoon (ISM) and the South East Asia Monsoon (SEAM). This study deals with a very important part of monsoon reconstructions as it focuses on changes in rainfall during the monsoon rather than changes in wind strength. Most proxies used for paleo-reconstructions are reconstructing changes in monsoon winds (e.g. Kroon et al., 1991; Gupta et al., 2015; Betzler et al., 2016), while changes in rainfall are difficult to reconstruct (e.g. Dettman et al., 2001; Huang et al., 2017). As such, this study would be a very good addition to

available literature. I was a bit confused by the “data-modelling” approach mentioned in the abstract, because this made me think new proxy data would be presented too. However, this leads to a final bit I am missing in the discussion on how these results support or counter evidence from proxy records. For several of the impacts you write that the literature is not conclusive, so could you now provide model evidence for a more likely scenario for Miocene changes in monsoon intensity?

We agree with the reviewer that the term ‘data-modelling approach’ is somewhat misleading. This was intended to refer to the boundary conditions (proxy data used for CO₂, ice volume, gateways and orogeny), but may have been read to imply data assimilation in the results themselves. We have replaced ‘data-modelling’ with ‘model-based’.

We have concluded that a sufficiently robust comparison with proxy data is beyond the scope of this paper, as detailed above in the response to Reviewer #1, but have provided a comparison with the simulations and semi-quantitative proxy data of Farnsworth et al (2019) in the response.

I have to point out that I am not a modeler myself, so I cannot judge if the model setup is correct or not. What I do wonder about though is how some of the sensitivity experiments are determined. Are these based on variable results from existing proxy reconstructions or rather to set a range in boundary conditions? For example closure of Panama at 5 Ma, while open is defined as 1000 m water depth. This is a very broad definition. The Montes papers claim closure by the mid-Miocene, which probably relates to deep water cutoff. But most of the literature on the direct effects of "effective" Panama closure (e.g. Haug and Tiedemann, 1998; Osborne et al., 2014; modelling studies by Haywood, Lunt, Dolan) use the interruption of surface water masses in the late Pliocene from 4 Ma to the NHG. This brings in a range of possibilities that could have a large impact on your results. Closing of Panama has often been associated to provide the settings for Northern Hemisphere Glaciation as a result of the cutoff of mixed layer water coming from the Pacific into the Atlantic.

Good point, and the reviewer is correct that the sensitivity to Panama closure is modelled as an instantaneous transition from 1200m gateway for computational tractability. We have added sensitivities to address this (see the new section “Boundary condition uncertainties” above). Firstly, we have simulated a transitional state, with a shallow (80m) seaway and secondly have tested the effect of uncertainty in the timing of closure (from 7Ma to 3Ma). These analysis reveal, as the reviewer suggests, that the transition is likely to have been substantially more complex than the simplified step function implied by the emulator.

In case of misunderstanding, we note that Northern Hemisphere Glaciation is not an emergent property of the model, but is an imposed boundary condition which is inferred from analysis (Stap et al 2017) of the benthic d18O record.

Similarly the final closure of the Tethys at the MCT, which some studies claim that TISW cessation may have triggered the MCT, but others say that the final closure was caused by the Antarctic glaciation through its sea level drop closing the final shallow bits of the Tethys, and thus that TISW already stopped much earlier. So the effect of a 1000 m deep gateway placed at the MCT or rather several million years earlier may make a large difference.

See previous response and the sensitivities to Panama closure in the new “Boundary condition uncertainties” section.

I suggest to have a look at the Raitzsch et al. (2020) paper and references in there dealing with atmospheric CO₂ concentrations from the MMCO crossing the MCT. The values in there are determined using B-isotopes in the shells of foraminifera. Based on these results a doubling of CO₂ in comparison with pre-industrial seems unlikely.

Thank you. We have added a sensitivity to address this, with CO₂ peaking at 560ppm at the MMCO, achieved this through a simple scaling function (see Fig S18 and “Boundary condition uncertainties” section).

An interesting point is that changes in the Hadley circulation seem mainly have an influence on the ISM, but not on the SEAM area. The discussion on explaining this impact seems to be mainly focused on factors related to changes in the northern hemisphere, e.g. ITCZ movement due to AMOC collapse or LGM ice sheets. But how realistic is this for the Miocene situation? I do not think that AMOC collapse was a big issue then, and ice sheet changes mainly occurred on Antarctica, which would then suggest a northward shift of the Westerlies and the ITCZ/Hadley circulation (Groeneveld et al., 2017).

We agree, that the effect of ice sheets is less relevant in the Miocene, and that other factors are dominating the changes in the Hadley circulation in this period. We refer to the existing text which discusses the roles of CO₂ and precession on the Hadley circulation over India:

Conversely, doubled CO₂ drives continental warming³⁸, which strengthens the Hadley circulation at 79°E and drives increased ISM rainfall. Similarly, a phase reversal in precession drives warming in the northern summer, a northerly shift of the JJA ITCZ, strengthened Hadley circulation and increased ISM rainfall.

How was the choice for the height of the HTP for specific time intervals determined, especially the interval 15-5 Ma? Despite the uncertainties most studies do seem to agree that the HTP is there by 15 Ma (see also Ding et al., 2017). How does a different scenario for this time interval change the results?

Tada et al. (2016) reviewed previous studies and concluded that there had been three major pulses of HTP uplift at c.40-35 Ma (south and central), 25-20 Ma (north) and 15-10 Ma (north-east and eastern). An Zhisheng et al. (2015) reviewed available research and conclude that, “by 35-20 Ma, the central Tibetan Plateau had been raised to 3000-4500m. The Tibetan Plateau then extended eastward and north-eastward, approaching present-day conditions, by 15-8 Mya. Further limited growth may have occurred to its northern, north-eastern and eastern margins since the Pliocene.” More recently Spicer et al (2020) have argued for a stepwise uplift pattern for Tibet involving several mountain ranges uplifted at different times with the plateau formation occurring during the late Neogene (< 7 Ma). So while it is the case that several reviews favour an uplifted and substantial Tibetan plateau by 15 Ma the various studies differ widely in their predictions. Moreover our results indicate that the elevation of the Tibetan plateau (as opposed to the Himalayan mountains) has little effect on the monsoon precipitation. The uplift histories of Southern Tibet (blue) and Himalayas (red) from Spicer et al (2020, Fig 9) , derived from fossilised leaf analysis, are copied below.

These scenario sensitivities are addressed in the “Boundary condition uncertainties” section. In particular, as the new text states The low-start linear orogeny approximates the growth of the Himalayas⁷⁵, included because of the sensitivity of the simulated Asian monsoon circulation to the Himalayan orogeny.

Minor comments:

Figure 1: please put both curves on their own y-axis.

Thanks, these are now added.

p.5 ref. 44: Kienast et al. Is cited here, but that seems wrong to me. This paper has nothing to do with the Pliocene closure of Panama.

Thanks, this was included in error and now removed.

Figure 2: why are the orogeny and gateway sensitivity records not continuing after 5 Ma?

The sensitivities do continue after 5Ma, but during this period the sensitivities take the same inputs as the baseline (modern orogeny and Panama closed), so the outputs are identical and the difference is zero.

To conclude, I think this manuscript would be a great addition for Nat. Comm. as it is dealing with a topic that currently receives a lot of attention, but I am missing some detail on the origin of the data used or rather on the different possibilities created by different datasets sometimes caused by missing literature. And an outlook or implications part at the end would allow the manuscript to be used for future proxy reconstructions.

We have added new text to the summary section

These results need to be validated in the proxy-based ISM precipitation records from the Miocene through to the Pliocene at orbital resolution. Multiple records capturing the spatial

and temporal variability in response to interacting climatic drivers will strengthen our understanding of the precipitation response.

Reviewers' Comments:

Reviewer #2:

Remarks to the Author:

This is my second review of the manuscript. The authors have extensively responded to the reviewer's comments and explained where they did follow and why, when they did not, follow the different issues. I only have a few more comments for consideration including one main point regarding a link to existing proxy-based reconstructions. After these points have been dealt with I see this manuscript as a valuable publication for Nature Communications.

"However, the interpretation of proxy records is complex, given dating uncertainties which significant on orbital time scales, differences between local-scale responses which proxy records may reflect (Nilsson-Kerr et al 2019), interactions between forcing mechanisms, and the limited number of records that directly capture precipitation. These problems also apply to the long-term monsoon evolution and the wide range of proxies which have been used are often contradictory in both magnitude and timing (Lunt et al 2010). A full model-data consensus reconstruction would be a major and highly challenging undertaking, well beyond the scope of this paper."

Although I do agree that a full data model comparison is beyond the scope of this study, I still think that this is an opportunity to point out these uncertainties and the many different interpretations existing on monsoon development for the late Cenozoic. Exactly this issue between proxies for wind strength vs proxies for precipitation is one of the main problems currently that is preventing consistent reconstructions. But also the large differences in long-term change considering the timing of monsoon intensification with some studies stating a strong monsoon since 20 Ma, others an intensification after the MMCT, and some as late as 12 Ma; and these differences are not due to age model errors. So by pointing this out and also referring to the sensitivity experiments you did, you could provide a very useful framework for future proxy-based studies.

Currently there are only two figures in the main text. I think it may be good to add the supplementary figure 1 to the main text as figure 2 to show what your results are based on. I would add the age intervals to the respective maps.

I suggest to make the y-axes in figure 2 cover the same range to emphasize the impact of different factors, e.g. the orogeny sensitivity now looks similar for both systems, but if you would put them both on the 0–1000 scale it becomes more obvious that the impact on the ISM is much lower.

Reference 5 is not complete.

Add spaces to many of your notations like 28mm > 28 mm, 30Ma > 30 Ma.

Reviewer #3 : As a “replacement reviewer” I will try to assess if the concerns raised by R#1 were addressed in the following questions/answers. All my comments will be in blue. I have added some personal remarks/questions at the end of R#1 section, along with the bibliographic references mentioned in my comments.

I find the authors produced a very substantial additional battery of tests to address the concerns of R#1 that add a great value to this study. Personally, I found this work very interesting, as it tries to consider all of the major forcings at stake throughout the Cenozoic, on different time scales (geological, orbital, etc.). The use of the PLASIM-GENIE EMIC and emulations allows to span a wide range of forcing combinations, which would be too time consuming for classic GCMs. The shortcoming of the method is of course the low resolution, which is fairly well addressed here. The paper might lack a bit of the paleoclimatic bibliography background about the state of the art regarding the evolution of the monsoons, especially for the older part (Oligocene), both from a conceptual point of view and from a data perspective.

My main concerns are:

- No mention of the impact of continentality increase (induced by the Paratethys and Tethys retreat) on Asian monsoons. This has been shown to be a key driver of monsoons evolution through time and, even if it wasn't tested here, it should be addressed
- The mixture of annual rainfall anomalies in square root mean or mm/yr (metrics of the paper) with JJA rainfall in mm/day (in several SI figures) hampers a clear understanding of the order of magnitude that are being compared. I feel the need for a clarification

Providing the few doubts raised here are addressed, we think this work would help all the paleoclimate community by offering a comprehensive view of the monsoon's evolution through the last 30 Ma.

Tectonic and climatic drivers of Asian monsoon evolution

James R. Thomson et al.

We are grateful for the detailed and constructive comments of both reviewers. We have now performed extensive additional analysis, as suggested by both, which has greatly strengthened the robustness of our conclusions. We have validated the simulated monsoon in both modern and paleoclimate states, quantified parametric uncertainty (model errors) through new simulation ensembles, tested simulated sensitivity to neglected and perturbed boundary conditions, and explored emulated uncertainty in response to the timings and magnitudes of boundary condition forcing. We regard a detailed comparison with proxies as outside of scope, as discussed below, but include a comparison with HadCM3 simulations and semi-quantitative proxy data (Farnsworth et al 2019).

Our responses are below, with reviewer comments in bold face, our responses in red type and revisions to the text in green. There are significant overlaps in the two reviews, so we have combined the responses, and most points are dealt with in the response to Reviewer #1.

Reviewer #1 (Remarks to the Author):

Thomson, et al. attempt to reconstruct the dominant forcing's and driving mechanisms behind the evolution of the Asian summer monsoon system, sub-divided into both the Indian and East Asian monsoon. As noted, it is difficult to ascertain the relative importance of different processes over geologic time which this work tries to address in a consistent manner. This is an interesting piece of work presented and confirms other recent studies on the various importance of certain boundary conditions on the response of the Asian monsoon system. I in particular like the attempt to investigate orbital variability as this is an arena that has seen far less attention over deep-time. I think this work has merit and eventually could be published; however, I have some concerns that would need to be addressed first.

Major comments:

The biggest concern is that this work is driven by a modern geography only. With paleogeography changing considerably over the last 30 million years, including (globally and regionally) oceanic gateways (e.g. Drake passage, Turgai strait, Panama isthmus) opening and closing, the closure and disappearance of the Paratethys, change in solar energy, ice sheets forming in the Southern and then Northern Hemisphere. While Thomson, et al. attempt to investigate some of these issues, they still use a modern geography and not those past time periods. Continents, mountain belts, deserts, ocean basins were all very different to that of today and would have a dissimilar impact on the Asian monsoon. This needs to be discussed more as these issues will have an impact on the thermodynamics of the monsoon system. Presently, this work mainly shows the impact of changing certain boundary conditions on the modern Asian monsoon system only.

Our approach, using intermediate complexity modelling and emulation, has enabled us to explore many drivers simultaneously, far more comprehensively than any previous study to our knowledge. We have included the most important drivers, including (from the reviewer's list) Panama closure, Tethys closure, ice-sheets in both hemispheres and evolving orogeny. Deserts are an emergent property of the vegetation model, and so these are also included. We have added sensitivity analysis (see below) that demonstrates solar variability is negligible with respect to the drivers we have included. We have also added new sensitivities to the Tibetan plateau and Himalayan orogeny, and to the bathymetry of the Panama gateway. We did not consider either the Drake passage opening or the Turgai strait closing as these events are generally accepted to have occurred before 30Ma, which made them low priority, and would have been major undertakings to simulate and emulate.

We have now added a new section on these uncertainties in the Methods:

Boundary condition uncertainties

Further uncertainties arise from the forcing boundary conditions. A similar simulation to our globally-scaled 30Ma orogeny was performed in HadCM3⁷¹, where global orogeny was reduced to 45% of modern relative to a world with a 1000m Tibetan plateau but otherwise flat. Runoff from the Yangtze and Pearl rivers, which have catchment areas in the north of our modelled SEAM region, was found to reduce to approximately half of that under present-day orogeny, consistent with the rainfall reductions seen in the emulator (Fig 2b) and the sensitivity simulation (Fig S3a). However, it is informative to examine the sensitivity with respect to the spatial distribution of the modelled orogeny. Fig S15 compares the distribution

of northern summer rainfall in the baseline preindustrial simulation with simulations that flatten either the Tibet Plateau or the Himalayas. Flattening the Tibet plateau has a modest effect on Asian monsoon rainfall, while flattening the Himalayas substantially reduces SEAM rainfall. This is consistent with earlier work⁷² which showed that the large-scale South Asian summer monsoon circulation is largely unaffected by removal of the Tibet plateau as long as the Himalayas are preserved. A closely related experiment⁷³, but at very high resolution ($0.23^\circ \times 0.31^\circ$), similarly concluded that the Himalayas are sufficient to sustain the Indian and Asian monsoon. High-resolution models are essential for detailed monsoon projections. For instance, 0.5° resolution has been found to be needed⁷⁴ to capture critical elements of the local monsoon moisture transport, a resolution which exceeds that of most state-of-the-art models. However, these comparisons suggest that large-scale monsoon features can be simulated at lower resolution, and furthermore that our emulators are unlikely to be sensitive to the details of the historic uplift.

A second important boundary condition uncertainty is the depth of Panama strait, which we have assumed to be $\sim 1000\text{m}$. We performed a simulation with an 80m deep strait, illustrated in Fig S16. In this simulation, freshening of the Atlantic does not penetrate to high northern latitudes and does not drive a weakening of the Atlantic meridional overturning circulation, nor of the Asian monsoon (in fact they both show increased strength, associated with increased surface Atlantic salinity at high latitudes). This illustrates, perhaps unsurprisingly, that the emulated step-change in ISM rainfall associated with Panama closure (Fig 2) is a highly simplified representation of the transition.

In Fig S17, we consider a 3.5Wm^{-2} reduction in the solar constant under 30Ma boundary conditions, an effect which was neglected in the emulators. Simulated annual SEAM rainfall is unchanged, ISM annual rainfall is reduced by 28mm , and there are changes in the seasonal distributions of both systems. However, these changes are modest (implied $\sqrt{\text{total effect}} \sim 10\text{mm}$) compared to the other drivers (Table 1).

A further source of boundary condition uncertainty, distinct from the implementation choices, are the uncertain timings and magnitudes of the forcings. We present emulator sensitivities in Figs S18, S19 and S20, which test the effects of Panama closure timing (7 to 3Ma), Tethys closure timing (17 to 13Ma), atmospheric CO_2 (reduced to 560ppm at the MMCO, see Fig S18) and changed orogeny timing (see Fig S18). The low-start linear orogeny approximates the growth of the Himalayas⁷⁵, included because of the sensitivity of the simulated Asian monsoon circulation to the Himalayan orogeny. These emulations do not show any unexpected behaviour, but serve to demonstrate the range of sensitivity to uncertain timing assumptions.

Regarding the sensitivity test on the removal of either Tibet or Himalayas (FigS15): the Himalayas are at most 1° wide and not strictly east-west oriented. A 5° resolution, like the one you are using, is too coarse to consider that the Himalayas only are flattened in your sensitivity experiment, it is more like the whole Southern Tibet region that was flattened here. The test strengthened the idea that it is the barrier effect of the southernmost orography that plays a crucial role on ISM formation rather than the eventual radiative effect of the plateau (Boos and Kuang 2010, Acosta and Huber 2020, Fluteau et al. 1999). It would be nice to simply nuance a bit the statement of “flattening the Himalayas”.

Model skill. How skillful is PLASIM-GENIE at reproducing the modern ISM and EASM statistics? It is essential to know so we can contextualize these results. An analysis of this would strengthen the conclusions of this ms.

Thanks, we agree, this is very helpful. We have added this (and other validations) in a new section in the Methods:

PLASIM-GENIE validation

The climate of PLASIM-GENIE has been validated against modern observations^{4,62} and against model inter-comparisons of the mid-Holocene, the Last Glacial Maximum, the Last Interglacial transient and the mid-Pliocene warm period⁶¹. These analyses, which all used the ‘optimised’ parameter set used to build the emulators, demonstrated that large-scale precipitation projections lie within the uncertainty envelope of high resolution IPCC-class models, which have themselves been validated against proxy data in the Mid-Holocene⁶⁶, the Last Glacial Maximum⁶⁶ and the Pliocene⁶⁷.

In this study, we have validated the simulated present-day monsoon system in our precise model setup in Fig S12, which compares global fields of precipitation (2005-2015) with NCEP reanalysis⁶⁸, and Fig S13 which compares the zonally averaged (70°E-90°E) distribution of modern simulated rainfall (2005-2015) through the seasonal cycle with both observations and high-resolution simulations⁶⁹. Zonally-averaged rainfall in PLASIM-GENIE peaks at 14mm/day, slightly later (August) and at slightly lower latitudes (15°N) than observations, but generally falling within the range of high-resolution model behaviour. We note that the emulator calculates annual average rainfall over large spatial scales (Fig 1) and therefore does not rely on the details of this spatiotemporal distribution.

Model resolution. Acosta and Huber (2020, 2017) (as well as others) have suggested that model resolution plays an important role in the ability to accurately simulate the characteristics of the Asian monsoon system. This needs further discussion and assessing the model skill wrt to the modern monsoon may help with this (if indeed the model does show skill).

Thanks again, addressed in the new “Boundary condition uncertainties” section above.

I may have missed it but I cannot see whether change in solar energy was incorporated into the emulation and GCMs? There would have been a 3-5 W/m² change over this 30 Ma. period. What influence might this have had?

Thanks, good point, also addressed in “Boundary condition uncertainties” above.

There is a lack of discussion on proxy-studies relating to orbital influences. I am left wondering whether the dominance of certain orbital signals is a result of the modern (although modified for sensitivity analysis) geography or whether these signals have been observed over the past 30 Ma. This would really add to the discussion and help contribute to the validity of the modelling results.

We are not aware of any proxy data at sufficient resolution to demonstrate orbital variability in the early record. However, the dominance of orbital influences is not the result of

geography in this model, and this is also now discussed within the new section on “PLASIM-GENIE parametric uncertainty” (see below) re the Eocene simulations of Keery et al (2018).

Such data are indeed rare, but they exist, for example in China for the latest Eocene/Oligocene in the Xining (Ao et al., 2020, Meijer et al., 2019) and Jiangnan basins (Huang et al., 2019). Note that these basins, located in distinct areas in China record either obliquity or precession preferentially. Moreover, the expression of the orbital forcing in the Xining basin changes after the EOT, suggesting possible high latitude remote influence at this ~40°N site. The Jiangnan basin, more to the South (~30°N), shows a more typical low latitude eccentricity/precession signal. This could be a good point to add to your discussion.

Model dependency. It is well known that different models can produce different results. While I would certainly not expect the authors to run the same set of simulations with different models, it would be nice to acknowledge that this. More so, it would also help comparing these results with other similar modelling work that is out there (e.g. Boos and Kuang, Acosta and Huber, Tardif, et al., Farnsworth, et al. 2019).

Another excellent point, which we have now addressed through parametric uncertainty. We have performed several perturbed parameter ensembles to address uncertainties in the model responses to the dominant drivers. These are detailed in a new section, pasted below.

PLASIM-GENIE parametric uncertainty

Model uncertainty is not captured by the emulator, because it was built from simulations with a single parameter set. In order to validate the robustness of the results under model uncertainty, we performed a series of perturbed parameter ensembles using a 69-member pre-calibrated parameter set⁶². These ensembles considered the dominant forcings (Table 1) of orogeny (scaling by 50%), CO₂ (doubling to 560ppm) and precession (reversed phase) and are summarised in Fig S14. Under parametric uncertainty, all three forcings drive ISM change that is significant with respect to the baseline preindustrial state and, despite significant uncertainty, all drive change of consistent sign. In all simulations, both doubled CO₂ and precession-reversal drive a strengthening of the monsoon while reduced orogeny drives a weakening. The optimised parameter set used to build the emulator lies close to the centre of all three ensembles. In contrast, and consistent with the emulator, simulated SEAM rainfall is dominated by orogeny in all ensemble members. SEAM changes driven by CO₂ and precession are modest and (in contrast to the ISM) of uncertain sign, indicating that the phasing of orbitally-driven SEAM change should be treated with caution. We note that precession was also found to be the dominant driver of Eocene monsoon variability simulated by PLASIM-GENIE in a fully altered paleogeography⁷⁰, thus the details of the geography do not appear to be important.

The “Boundary condition uncertainties” section above includes comparisons with related modelling work as suggested (Boos and Kuang 2010, Lunt et al 2010, Acosta and Huber 2017, Acosta and Huber 2020, Keery et al 2018).

As mentioned by R#1 we believe all of the suggested articles are equally necessary citations. **Tardif et al. 2020** discuss in depth the influence of some key paleogeographic aspects on Asian monsoonal climates.

Proxy data. The real weakness of this study is a lack of Proxy data. I would like to see some attempt to evaluate the response suggested by this emulation and GCM approach. Presently it is hard to determine how much confidence we can have in these results when it is difficult to see whether the model and emulator are accurately able to reconstruct past time periods and change (which will be difficult given the modern, but modified land geography). There are a host of palaeoflora and biogeochemical evidence out there to do this. Again, addition and discussion of this would help justify these conclusions.

The general conclusions of our emulations are consistent with the general inferences from proxy data, such as a strengthening through time, presumed in response to orogeny, and variability at orbital time scales that is driven, whether directly or indirectly (e.g. ice sheets and CO₂), by precession and obliquity (An Zhisheng et al 2015). However, the interpretation of proxy records is complex, given dating uncertainties which significant on orbital time scales, differences between local-scale responses which proxy records may reflect (Nilsson-Kerr et al 2019), interactions between forcing mechanisms, and the limited number of records that directly capture precipitation. These problems also apply to the long-term monsoon evolution and the wide range of proxies which have been used are often contradictory in both magnitude and timing (Lunt et al 2010). A full model-data consensus reconstruction would be a major and highly challenging undertaking, well beyond the scope of this paper. However, we have plotted a comparison below, of our SEAM rainfall emulations averaged over 1Ma time intervals (superposed orange circles in lower panel) against the simulations (lower) and semi-quantitative proxy data (upper) in Farnsworth et al (2019). The comparison shows remarkable similarity in precipitation trend for SEAM.

An Zhisheng et al (2015) Global monsoon dynamics and climate change, *Annu. Rev. Earth Planet. Sci.* 43, 29-77

K. Nilsson-Kerr, P. Anand, P. F. Sexton, M. J. Leng, S. Misra, S. C. Clemens, S. J. Hammond, Role of Asian summer monsoon subsystems in the inter-hemispheric progression of deglaciation. *Nat. Geosci.* 12, 290 (2019).

Lunt, D.J., Flecker, R. and Clift, P.D. (2010) The impacts of Tibetan uplift on paleoclimate proxies, in Clift, P. D., Tada, R. & Zheng, H. (eds) *Monsoon Evolution and Tectonics–Climate Linkage in Asia*. Geological Society, London, Special Publications, 342, 279–291.

Very interesting comparison with Farnsworth et al. 2019, any idea why the last 4 Ma or so don't match at all, while the 30-5Ma records are very similar?

If the comparison with all available data would indeed be time consuming, a few papers propose a sum-up of the Asian monsoons evolution during your period of interest. You already cite some of them, but do not use them to their full potential.

Wang et al (2005) offer a wide view of the forcings at stakes on the monsoon phenomenon, while Tada et al. (2007) propose a view of both ISM and SEAM evolution with respect to Himalayan and Tibetan orogeny (Fig 7 in particular). Molnar et al. (2010) also propose a rather comprehensive view of the major monsoon indicators available in Asia since the last 20 Ma. If their evolution is usually not precise up to the orbital time-scale, they should be

used to validate and/or discuss the longer time-scale pattern of amplification you are simulating and the remaining uncertainties.

Tectonic changes. There seems to my knowledge be two competing tectonic considerations being presented with regards to Tibetan orogeny and evolution through the Cenozoic. Botsyun, et al. use a new and innovative approach to suggest a ‘low’ plateau rising while Spicer, et al. suggest two large mountain ranges with a low valley in between infilling. I am also not sure the reference cited does suggest that “modelled rainfall is insensitive to the details of the paleogeographic reconstruction”, from looking at this reference they show the opposite and that CO₂ was not a big factor. Some discussion on the implications of resolution and the potential impacts of different HTP should be discussed. More so, Acosta and Huber also show that while a monsoon will always be present regardless of the addition of Tibet or not, the details do make a huge difference in the direction of moist flow in the Asian monsoon region. This brings me back to the resolution issue of the GENIE. Acosta and Huber (2017, 2020) show that you cannot get a realistic monsoon developing using current low-resolution GCMs let alone even coarser intermediate models such as GENIE. This needs to be discussed in some manner in this manuscript.

Thanks again, and also addressed in new section on “Boundary condition uncertainties”. We have removed the sentence “In addition, some recent simulation work has shown that modelled rainfall is fairly insensitive to the details of paleogeographic reconstruction.”

As mentioned by R#1 we believe **Botsyun et al. 2019** should be cited as well (Line 176), given they represent the main alternative hypothesis to paleoaltimetry-based methods from Rowley et al. and Spicer et al.

References:

R.P. Acosta, M. Huber. (2020). Competing Topographic Mechanisms for the Summer Indo-Asian Monsoon Geophysical Research Letters, 47

Acosta, R. P., & Huber, M. (2017). The neglected Indo-Gangetic Plains low-level jet and its importance for moisture transport and precipitation during the peak summer monsoon. Geophysical Research Letters, 44, 8601– 8610

Boos, W., & Kuang, Z. (2010). Dominant control of the South Asian monsoon by orographic insulation versus plateau heating. Nature.

S. Botsyun, P. Sepulchre, Y. Donnadieu, C. Risi, A. Licht, J. K. Caves Rügenstein. (2019). Revised paleoaltimetry data show low Tibetan Plateau elevation during the Eocene. Science 363.

Delphine Tardif, Frédéric Fluteau, Yannick Donnadieu, Guillaume Le Hir, Jean-Baptiste Ladant, et al. 2020. The origin of Asian monsoons: a modelling perspective. Climate of the Past, 16 (3), pp.847-865

Additional questions from R#3 to the authors:

On the manuscript

Line 17: “multiple drivers of Asian monsoon variability: orbital forcing, atmospheric carbon dioxide (CO₂), global ice volume, Himalayan-Tibetan Plateau (HTP) uplift, tectonically-induced changes to major ocean gateways, and Intertropical Convergence Zone (ITCZ) movement, via atmospheric circulation changes and Atlantic Meridional Ocean Circulation (AMOC) teleconnections »

R#3 You mention here and further analyze the closing of the Tethys, but totally overlook the impact of continentality changes induced by the retreat of the Paratethys, initiating in the Eocene and continuing during the Oligocene and the Miocene (Kaya et al. 2019). Studies have shown that the increased continentality over the progressively emerging Anatolia, Arabia, North Africa and central Europe (induced by the Tethys and Paratethys retreat due to global eustasy and to the convergence of Africa and Eurasia) do trigger changes on the monsoon strength (Zhang et al. 2007, Ramstein et al. 1997, Fluteau et al. 1999). It is mostly due to a displacement of main low and high sea level pressure patterns over the continents, that drive increased moisture convergence over Asia. This is also discussed in Tardif et al. 2020.

Although testing this would require more refined paleogeography reconstructions, which is out of the scope here, I think it is paramount to mention it, in the introduction and also in the results regarding the Tethys, since it is to me the major (only) point that your study hasn't considered (line ~133 for ex).

Line 126:

R#3 I find generally confusing that most conclusions in the text are made considering the annual rainfall changes induced by changing parameters, whereas the corresponding figures display seasonal JJA precipitation changes (Fig. S9, S17). You state that SEAM changes significantly under cross-interaction between forcings (Line 126: from 1320 to 1300 mm and from 1216 to 1242 mm). This represent changes of ~20-26 mm/yr that you find significant. However, when testing the solar constant reduction (see further comment for Line 330), the change of ~28 mm/yr on the ISM is considered not significant. I understand that, given the wider variability in annual ISM rainfall, this change induced by solar constant might indeed be of second order. However, you use the figures S9 and S17 to support these conclusions, whereas these figures show JJA precipitations anomalies that are both in the order of +/- 1mm in the ISM domain. I find this mixture in seasonal versus annual rainfall blurs the message. I think the Supp Figures should at least display the anomalies mentioned in the text (annual rainfall values). Moreover, you dismiss the effect of the solar constant based on its minimal square root value compared to other values showed in Table 1. This add a third way to explain a single variability. You might pick one that is understandable (square root, annual or seasonal anomaly) or show all.

Lines 136, 140:

R#3 You refer to a « compression » of the ITCZ due to ice sheet buildup, that « strengthens the tropical hydrological cycle ». I don't find this statement very clear from a physical or mechanistic point of view: does it mean the precipitations quantities are unchanged but concentrated over a narrower band? that the moisture convergence is increased? that the wind speed increases?

Line 150: « We infer this ISM rainfall increase to be driven by indirect teleconnections arising from constriction of the Panama gateway, which led to increased AMOC²⁹ (Supplementary Fig.6) and consequent northward movement of the ITCZ. »

R#3 What would be the link between an increased AMOC and a northward migration of the ITCZ? are there modelling studies or proxies allowing to make this assumption?

Line 300: “We note that precession was also found to be the dominant driver of Eocene monsoon variability simulated by PLASIM-GENIE in a fully altered paleogeography⁷⁰, thus the details of the geography do not appear to be important »

R#3 This statement is problematic to me because these forcings don't act on the same timescales. Precession is the dominant mode of the monsoon variability on time scales of several thousands of years. On longer timescales, paleogeography becomes the dominant mode. Moreover, as you average your results over large regions and with low spatial resolution, you cannot assume the results would be strictly similar in a finer resolved model. Please consider removing “thus the details of the geography do not appear to be important”

Line 330 (Fig S17): « Simulated annual SEAM rainfall is unchanged, ISM annual rainfall is reduced by 28mm, and there are changes in the seasonal distributions of both systems. However, these changes are modest (implied ν total effect ~ 10mm) compared to the other drivers (Table 1). »

R#3 Could you add the anomaly plots on the annual rainfall too? Otherwise, it seems weird to say that SEAM annual rainfall is unchanged but showing in FigS17 that summer precipitation is significantly increased in this region. Your paper investigates regional summer monsoons, and your figure actually shows an increase in summer SEAM precipitation due to increase solar constant. If you decide to base your arguments regarding monsoon strengthening on the annual rainfall metric (like Farnsworth et al.), then showing this annual anomaly in S17 seems to better serve your purpose.

Table 1

R#3 : the 3 central columns referring to the Miocene are unclear to me. Is the "Miocene" column somehow a weighted average of the values obtained for Late and early Miocene? Given that the impact of obliquity on the “Miocene” is higher (17) than for the early (15) and the late (16) Miocene, I am confused.

You mention that more details are available in Supp Material 1.2, but this section doesn't appear clearly in the supplementary file I have.

On the Supplementary Information:

FigS14:

R#3 The legend of the figure should be made clearer. What are the values in y axes (mm I guess)? What is the legend of the color bars (black, orange, yellow), mean, median and 5-95% interval?

Additional references mentioned in my comments:

Ao, H., Dupont-Nivet, G., Rohling, E. J., Zhang, P., Ladant, J.-B., Roberts, A. P., Licht, A., Liu, Q., Liu, Z., Dekkers, M. J., Coxall, H. K., Jin, Z., Huang, C., Xiao, G., Poulsen, C. J., Barbolini, N., Meijer, N., Sun, Q., Qiang, X., Yao, J. and An, Z.: Orbital climate variability on the northeastern Tibetan Plateau across the Eocene–Oligocene transition, *Nature Communications*, 11(1), doi:[10.1038/s41467-020-18824-8](https://doi.org/10.1038/s41467-020-18824-8), 2020.

Meijer, N., Dupont-Nivet, G., Abels, H. A., Kaya, M. Y., Licht, A., Xiao, M., Zhang, Y., Roperch, P., Poujol, M., Lai, Z. and Guo, Z.: Central Asian moisture modulated by proto-Paratethys Sea incursions since the early Eocene, *Earth and Planetary Science Letters*, 510, 73–84, doi:[10.1016/j.epsl.2018.12.031](https://doi.org/10.1016/j.epsl.2018.12.031), 2019.

Huang, C. and Hinnov, L.: Astronomically forced climate evolution in a saline lake record of the middle Eocene to Oligocene, Jiangnan Basin, China, *Earth and Planetary Science Letters*, 528, 115846, doi:[10.1016/j.epsl.2019.115846](https://doi.org/10.1016/j.epsl.2019.115846), 2019.

Molnar, P., Boos, W. R. and Battisti, D. S.: Orographic Controls on Climate and Paleoclimate of Asia: Thermal and Mechanical Roles for the Tibetan Plateau, *Annual Review of Earth and Planetary Sciences*, 38(1), 77–102, doi:[10.1146/annurev-earth-040809-152456](https://doi.org/10.1146/annurev-earth-040809-152456), 2010.

Kaya, M. Y., Dupont-Nivet, G., Proust, J., Roperch, P., Bougeois, L., Meijer, N., Frieling, J., Fioroni, C., Özkan Altıner, S., Vardar, E., Barbolini, N., Stoica, M., Aminov, J., Mamtimin, M. and Zhaojie, G.: Paleogene evolution and demise of the proto-Paratethys Sea in Central Asia (Tarim and Tajik basins): Role of intensified tectonic activity at ca. 41 Ma, *Basin Research*, 31(3), 461–486, doi:[10.1111/bre.12330](https://doi.org/10.1111/bre.12330), 2019.

Zhang, Z., Wang, H., Guo, Z. and Jiang, D.: What triggers the transition of palaeoenvironmental patterns in China, the Tibetan Plateau uplift or the Paratethys Sea retreat?, *Palaeogeography, Palaeoclimatology, Palaeoecology*, 245(3–4), 317–331, doi:[10.1016/j.palaeo.2006.08.003](https://doi.org/10.1016/j.palaeo.2006.08.003), 2007.

Ramstein, G., Fluteau, F., Besse, J. and Joussaume, S.: Effect of orogeny, plate motion and land–sea distribution on Eurasian climate change over the past 30 million years, *Nature*, 386(6627), 788–795, doi:[10.1038/386788a0](https://doi.org/10.1038/386788a0), 1997.

Fluteau, F., Ramstein, G. and Besse, J.: Simulating the evolution of the Asian and African monsoons during the past 30 Myr using an atmospheric general circulation model, *Journal of Geophysical Research: Atmospheres*, 104(D10), 11995–12018, doi:[10.1029/1999JD900048](https://doi.org/10.1029/1999JD900048), 1999.

Braconnot, P., Joussaume, S., Marti, O. and de Noblet, N.: Synergistic feedbacks from ocean and vegetation on the African Monsoon response to Mid-Holocene insolation, *Geophysical Research Letters*, 26(16), 2481–2484, doi:[10.1029/1999GL006047](https://doi.org/10.1029/1999GL006047), 1999.

Liu, X., Kutzbach, J. E., Liu, Z., An, Z. and Li, L.: The Tibetan Plateau as amplifier of orbital-scale variability of the East Asian monsoon: TIBET PLATEAU AND MONSOON VARIABILITY, *Geophysical Research Letters*, 30(16), doi:[10.1029/2003GL017510](https://doi.org/10.1029/2003GL017510), 2003.

Reviewer #2 (Remarks to the Author):

This is a review of the manuscript „Tectonic and climatic drivers of Asian monsoon evolution“ by Thomson et al. The manuscript investigates how different impacts, ice sheets, tectonics, CO₂, orbital variability, have influenced changes in monsoon rainfall for the last 30 Myr using a model setup allowing to distinguish the impact of each separate factor. The main focus is on changes in the Indian Summer Monsoon (ISM) and the South East Asia Monsoon (SEAM). This study deals with a very important part of monsoon reconstructions as it focuses on changes in rainfall during the monsoon rather than changes in wind strength. Most proxies used for paleo-reconstructions are reconstructing changes in monsoon winds (e.g. Kroon et al., 1991; Gupta et al., 2015; Betzler et al., 2016), while changes in rainfall are difficult to reconstruct (e.g. Dettman et al., 2001; Huang et al., 2017). As such, this study would be a very good addition to available literature. I was a bit confused by the “data-modelling” approach mentioned in the abstract, because this made me think new proxy data would be presented too. However, this leads to a final bit I am missing in the discussion on how these results support or counter evidence from proxy records. For several of the impacts you write that the literature is not conclusive, so could you now provide model evidence for a more likely scenario for Miocene changes in monsoon intensity?

We agree with the reviewer that the term ‘data-modelling approach’ is somewhat misleading. This was intended to refer to the boundary conditions (proxy data used for CO₂, ice volume, gateways and orogeny), but may have been read to imply data assimilation in the results themselves. We have replaced ‘data-modelling’ with ‘model-based’.

We have concluded that a sufficiently robust comparison with proxy data is beyond the scope of this paper, as detailed above in the response to Reviewer #1, but have provided a comparison with the simulations and semi-quantitative proxy data of Farnsworth et al (2019) in the response.

I have to point out that I am not a modeler myself, so I cannot judge if the model setup is correct or not. What I do wonder about though is how some of the sensitivity experiments are determined. Are these based on variable results from existing proxy reconstructions or rather to set a range in boundary conditions? For example closure of Panama at 5 Ma, while open is defined as 1000 m water depth. This is a very broad definition. The Montes papers claim closure by the mid-Miocene, which probably relates to deep water cutoff. But most of the literature on the direct effects of "effective" Panama closure (e.g. Haug and Tiedemann, 1998; Osborne et al., 2014; modelling studies by Haywood, Lunt, Dolan) use the interruption of surface water masses in the late Pliocene from 4 Ma to the NHG. This brings in a range of possibilities that could have a large impact on your results. Closing of Panama has often been associated to provide the settings for Northern Hemisphere Glaciation as a result of the cutoff of mixed layer water coming from the Pacific into the Atlantic.

Good point, and the reviewer is correct that the sensitivity to Panama closure is modelled as an instantaneous transition from 1200m gateway for computational tractability. We have added sensitivities to address this (see the new section “Boundary condition uncertainties” above). Firstly, we have simulated a transitional state, with a shallow (80m) seaway and secondly have tested the effect of uncertainty in the timing of closure (from 7Ma to 3Ma).

These analysis reveal, as the reviewer suggests, that the transition is likely to have been substantially more complex than the simplified step function implied by the emulator.

In case of misunderstanding, we note that Northern Hemisphere Glaciation is not an emergent property of the model, but is an imposed boundary condition which is inferred from analysis (Stap et al 2017) of the benthic $\delta^{18}\text{O}$ record.

Similarly the final closure of the Tethys at the MCT, which some studies claim that TISW cessation may have triggered the MCT, but others say that the final closure was caused by the Antarctic glaciation through its sea level drop closing the final shallow bits of the Tethys, and thus that TISW already stopped much earlier. So the effect of a 1000 m deep gateway placed at the MCT or rather several million years earlier may make a large difference.

See previous response and the sensitivities to Panama closure in the new “Boundary condition uncertainties” section.

I suggest to have a look at the Raitzsch et al. (2020) paper and references in there dealing with atmospheric CO_2 concentrations from the MMCO crossing the MCT. The values in there are determined using B-isotopes in the shells of foraminifera. Based on these results a doubling of CO_2 in comparison with pre-industrial seems unlikely.

Thank you. We have added a sensitivity to address this, with CO_2 peaking at 560ppm at the MMCO, achieved this through a simple scaling function (see Fig S18 and “Boundary condition uncertainties” section).

An interesting point is that changes in the Hadley circulation seem mainly have an influence on the ISM, but not on the SEAM area. The discussion on explaining this impact seems to be mainly focused on factors related to changes in the northern hemisphere, e.g. ITCZ movement due to AMOC collapse or LGM ice sheets. But how realistic is this for the Miocene situation? I do not think that AMOC collapse was a big issue then, and ice sheet changes mainly occurred on Antarctica, which would then suggest a northward shift of the Westerlies and the ITCZ/Hadley circulation (Groeneveld et al., 2017).

We agree, that the effect of ice sheets is less relevant in the Miocene, and that other factors are dominating the changes in the Hadley circulation in this period. We refer to the existing text which discusses the roles of CO_2 and precession on the Hadley circulation over India:

Conversely, doubled CO_2 drives continental warming³⁸, which strengthens the Hadley circulation at 79°E and drives increased ISM rainfall. Similarly, a phase reversal in precession drives warming in the northern summer, a northerly shift of the JJA ITCZ, strengthened Hadley circulation and increased ISM rainfall.

How was the choice for the height of the HTP for specific time intervals determined, especially the interval 15-5 Ma? Despite the uncertainties most studies do seem to agree that the HTP is there by 15 Ma (see also Ding et al., 2017). How does a different scenario for this time interval change the results?

Tada et al. (2016) reviewed previous studies and concluded that there had been three major pulses of HTP uplift at c.40-35 Ma (south and central), 25-20 Ma (north) and 15-10 Ma (north-east and eastern). An Zhisheng et al. (2015) reviewed available research and conclude that, “by 35-20 Ma, the central Tibetan Plateau had been raised to 3000-4500m. The Tibetan Plateau then extended eastward and north-eastward, approaching present-day conditions, by 15-8 Mya. Further limited growth may have occurred to its northern, north-eastern and eastern margins since the Pliocene.” More recently Spicer et al (2020) have argued for a stepwise uplift pattern for Tibet involving several mountain ranges uplifted at different times with the plateau formation occurring during the late Neogene (< 7 Ma). So while it is the case that several reviews favour an uplifted and substantial Tibetan plateau by 15 Ma the various studies differ widely in their predictions. Moreover our results indicate that the elevation of the Tibetan plateau (as opposed to the Himalayan mountains) has little effect on the monsoon precipitation. The uplift histories of Southern Tibet (blue) and Himalayas (red) from Spicer et al (2020, Fig 9) , derived from fossilised leaf analysis, are copied below.

These scenario sensitivities are addressed in the “Boundary condition uncertainties” section. In particular, as the new text states The low-start linear orogeny approximates the growth of the Himalayas⁷⁵, included because of the sensitivity of the simulated Asian monsoon circulation to the Himalayan orogeny.

Minor comments:

Figure 1: please put both curves on their own y-axis.

Thanks, these are now added.

p.5 ref. 44: Kienast et al. Is cited here, but that seems wrong to me. This paper has nothing to do with the Pliocene closure of Panama.

Thanks, this was included in error and now removed.

Figure 2: why are the orogeny and gateway sensitivity records not continuing after 5 Ma?

The sensitivities do continue after 5Ma, but during this period the sensitivities take the same inputs as the baseline (modern orogeny and Panama closed), so the outputs are identical and the difference is zero.

To conclude, I think this manuscript would be a great addition for Nat. Comm. as it is dealing with a topic that currently receives a lot of attention, but I am missing some detail on the origin of the data used or rather on the different possibilities created by different datasets sometimes caused by missing literature. And an outlook or implications part at the end would allow the manuscript to be used for future proxy reconstructions.

We have added new text to the summary section

These results need to be validated in the proxy-based ISM precipitation records from the Miocene through to the Pliocene at orbital resolution. Multiple records capturing the spatial and temporal variability in response to interacting climatic drivers will strengthen our understanding of the precipitation response.

Tectonic and climatic drivers of Asian monsoon evolution

James R. Thomson et al.

We thank both reviewers, and especially note the very useful comments from the new reviewer #3 from a modelling perspective. Our responses are below, with reviewer comments in bold face, our responses in red type and revisions to the text in green.

During checking we identified a minor error in the total effect calculation which we have corrected, revising Table 1 accordingly. The changes needed to the text are minimal, highlighted orange in the revised manuscript. Most notably, the relative contributions to 30Ma variance have changed from 42% to 39% (orogeny), from 24% to 25% (precession) and from 20% to 21% (CO₂). We have now also calculated uncertainties for total effects, included in the Table and explained in the caption “Mean and 1-sigma uncertainties are provided for each by repeating each calculation ten times with different (stochastically derived) emulators”.

Reviewer #2 (Remarks to the Author):

This is my second review of the manuscript. The authors have extensively responded to the reviewer’s comments and explained where they did follow and why, when they did not, follow the different issues. I only have a few more comments for consideration including one main point regarding a link to existing proxy-based reconstructions. After these points have been dealt with I see this manuscript as a valuable publication for Nature Communications.

“However, the interpretation of proxy records is complex, given dating uncertainties which significant on orbital time scales, differences between local-scale responses which proxy records may reflect (Nilsson-Kerr et al 2019), interactions between forcing mechanisms, and the limited number of records that directly capture precipitation. These problems also apply to the long-term monsoon evolution and the wide range of proxies which have been used are often contradictory in both magnitude and timing (Lunt et al 2010). A full model-data consensus reconstruction would be a major and highly challenging undertaking, well beyond the scope of this paper.”

Although I do agree that a full data model comparison is beyond the scope of this study, I still think that this is an opportunity to point out these uncertainties and the many different interpretations existing on monsoon development for the late Cenozoic. Exactly this issue between proxies for wind strength vs proxies for precipitation is one of the main problems currently that is preventing consistent reconstructions. But also the large differences in long-term change considering the timing of monsoon intensification with some studies stating a strong monsoon since 20 Ma, others an intensification after the MMCT, and some as late as 12 Ma; and these differences are not due to age model errors. So by pointing this out and also referring to the sensitivity experiments you did, you could provide a very useful framework for future proxy-based studies.

We have added a discussion on data model comparisons:

It is instructive to compare these long-term changes with a review of observational evidence³⁴ that concluded that the SEAM intensified between 25 and 20 Ma, putatively in response to northern HTP uplift. Further intensification has been associated with northeastern and

eastern HTP uplift from 15 to 10 Ma^{35,36}, possibly masked by the competing effects of CO₂ drawdown, cooling and increasing glaciation³⁴. Our emulations are broadly consistent with these inferences, although orographic forcing dominates in our results, and clear net strengthening of SEAM rainfall is emulated in response. Proxy evidence further indicates transient increases in SEAM intensity at around 16-14 Ma and 4.2–2.8 Ma³⁴ where orography is dominant in our SEAM emulation. However, we do emulate an intensification in ISM rainfall around these times (14-12 Ma and 4-3 Ma). During the Pleistocene, proxies indicate SEAM weakening, but with increased orbital variability³⁴. This weakening is apparent (though modest) in the SEAM emulation, driven by a combination of ice-sheet and CO₂ forcing. Although increased variability is evident in our emulated records, this increase occurs earlier, over the period 15-13 Ma, possibly associated with the final HTP uplift, unrelated to Pleistocene boundary conditions. Orbital timescale monsoon variability has been evidenced in proxy records throughout the last 30 Ma and earlier^{37,38,39}, suggestive of both low-latitude precessional forcing and high-latitude obliquity forcing via ice sheets, and consistent with our emulators. A direct comparison of orbital variability between our regional-scale emulators and individual, locally-influenced proxy records would be extremely challenging⁴⁰, but possible in principle if a sufficient density of observational records could be assembled.

Currently there are only two figures in the main text. I think it may be good to add the supplementary figure 1 to the main text as figure 2 to show what your results are based on. I would add the age intervals to the respective maps.

Thanks, good ideas, we have moved and edited as suggested.

I suggest to make the y-axes in figure 2 cover the same range to emphasize the impact of different factors, e.g. the orogeny sensitivity now looks similar for both systems, but if you would put them both on the 0—1000 scale it becomes more obvious that the impact on the ISM is much lower.

We prefer to leave the ranges unchanged as the underlying patterns of change are clearer. We have added text to emphasise these ranges are different.

Note the different y-axis scales used for the different forcings and for the two monsoon systems.

Reference 5 is not complete.

We have added the missing publication year

Add spaces to many of your notations like 28mm > 28 mm, 30Ma > 30 Ma.

Done

Reviewer #3

As a “replacement reviewer” I will try to assess if the concerns raised by R#1 were addressed in the following questions/answers. All my comments will be in blue. I have added some personal remarks/questions at the end of R#1 section, along with the bibliographic references mentioned in my comments.

I find the authors produced a very substantial additional battery of tests to address the concerns of R#1 that add a great value to this study. Personally, I found this work very interesting, as it tries to consider all of the major forcings at stake throughout the Cenozoic, on different time scales (geological, orbital, etc.). The use of the PLASIM-GENIE EMIC and emulations allows to span a wide range of forcing combinations, which would be too time consuming for classic GCMs. The shortcoming of the method is of course the low resolution, which is fairly well addressed here. The paper might lack a bit of the paleoclimatic bibliography background about the state of the art regarding the evolution of the monsoons, especially for the older part (Oligocene), both from a conceptual point of view and from a data perspective.

My main concerns are:

- - No mention of the impact of continentality increase (induced by the Paratethys and Tethys retreat) on Asian monsoons. This has been shown to be a key driver of monsoons evolution through time and, even if it wasn't tested here, it should be addressed

Many thanks we have added text to the introduction:

By simulating three alternative paleogeographies, we capture the first-order effects of the two main ocean gateways, but not the full effects of the retreat of the Tethys and Paratethys oceans¹¹⁻¹³, still ongoing in the Miocene¹⁴, with increasing continentality displacing pressure systems and triggering changes in monsoon strength.

- - The mixture of annual rainfall anomalies in square root mean or mm/yr (metrics of the paper) with JJA rainfall in mm/day (in several SI figures) hampers a clear understanding of the order of magnitude that are being compared. I feel the need for a clarification

The square root (total effect) converts the total effect to units of mm/yr, noting that total effects are a variance measure and the transformation converts them to standard deviations. We have added text in the caption to clarify this:

Total effects (aside from percentages) are provided as their square roots to aid comparison by converting this variance measure to units of mm/day.

With respect to annual anomalies, we have added new text (introduction to supplementary figures)

Sensitivities are presented as Jun-July-August averages to highlight the changes in monsoon dynamics. Exceptions are the model sensitivities S16 and S17, which illustrate the change in annual rainfall for direct comparison with the emulated quantities.

a new panel to S16 (with annual rainfall)

Reduced solar constant (-3.5Wm^{-2}) at 30MA.

Supplementary Fig.16: Changes in a) Jun-Jul-Aug rainfall, b) Jun-Jul-Aug surface air temperature and c) annual average rainfall in response to a 3.5Wm^{-2} reduction in the solar constant under 30Ma boundary conditions. Note the legend for annual rainfall (± 200 mm), which is an order of magnitude lower than the range used for the dominant drivers (Supplementary Fig. 17).

and a new figure S17, which plots the five main sensitivities as annual rainfall for direct comparison with the emulators. These figures are referred to in the text where relevant.

Supplementary Fig.17: PLASIM-GENIE sensitivity simulations for change relative to preindustrial conditions, plotting annual precipitation change for (a) 50% orography, (b) doubled CO_2 , (c) LGM ice sheet, (d) open Panama, (e) reversed precession phase.

Providing the few doubts raised here are addressed, we think this work would help all the paleoclimate community by offering a comprehensive view of the monsoon's evolution through the last 30 Ma.

Re R#1 response

Regarding the sensitivity test on the removal of either Tibet or Himalayas (FigS15): the Himalayas are at most 1° wide and not strictly east-west oriented. A 5° resolution, like the one you are using, is too coarse to consider that the Himalayas only are flattened in your sensitivity experiment, it is more like the whole Southern Tibet region that was flattened here. The test strengthened the idea that it is the barrier effect of the southernmost orography that plays a crucial role on ISM formation rather than the eventual radiative effect of the plateau (Boos and Kuang 2010, Acosta and Huber 2020, Fluteau et al. 1999). It would be nice to simply nuance a bit the statement of “flattening the Himalayas”.

Thanks, we have revised to

Supplementary Fig. 14 compares the distribution of northern summer rainfall in the baseline preindustrial simulation with simulations that flatten either the northern Tibetan Plateau or the southern Tibetan plateau, which includes the Himalayas. Flattening the northern plateau has a modest effects on both systems, while flattening the southern plateau and Himalayas substantially reduces SEAM rainfall.

[Data showing orbital variability in the early record] are indeed rare, but they exist, for example in China for the latest Eocene/Oligocene in the Xining (Ao et al., 2020, Meijer et al., 2019) and Jiangnan basins (Huang et al., 2019). Note that these basins, located in distinct areas in China record either obliquity or precession preferentially. Moreover, the expression of the orbital forcing in the Xining basin changes after the EOT, suggesting possible high latitude remote influence at this ~40°N site. The Jiangnan basin, more to the South (~30°N), shows a more typical low latitude eccentricity/precession signal. This could be a good point to add to your discussion.

Thank you, we have included these references in the new discussion on proxies (see above), specifically

Orbital timescale monsoon variability has been evidenced in proxy records throughout the last 30 Ma and earlier^{37,38,39}, suggestive of both low-latitude precessional forcing and high-latitude obliquity forcing via ice sheets, and consistent with our emulators.

As mentioned by R#1 we believe all of the suggested articles are equally necessary citations. **Tardif et al. 2020** discuss in depth the influence of some key paleogeographic aspects on Asian monsoonal climates.

We have added the following:

We note that simulations⁸⁷ in a 40 Ma paleogeography (pre-dating the assumed range of validity of our emulator), inferred substantially altered atmospheric circulation patterns from modern conditions, driving aridity over northern India and the HTP. While we capture elements of this in our scaled orography experiments (Supplementary Figs. 2a and 17a), the

drying in PLASIM-GENIE is less pronounced and located further east. These differences may, at least in part, reflect the different orography assumptions.

Very interesting comparison with Farnsworth et al. 2019, any idea why the last 4 Ma or so don't match at all, while the 30-5Ma records are very similar?

We apply a global scaling of orography (that reaches present-day at 5 Ma and remains there). In contrast, Farnsworth et al 2019 use different paleogeographies for each snapshot simulation. The mean elevation of these orographies reduces from ~2750 m at ~5 Ma to ~2400 m at ~3 Ma (diamonds in Fig 1c of Farnsworth et al 2019, pasted below), and this is likely driving a reduction in simulated precipitation that is not forced in our emulator.

If the comparison with all available data would indeed be time consuming, a few papers propose a sum-up of the Asian monsoons evolution during your period of interest. You already cite some of them, but do not use them to their full potential. Wang et al (2005) offer a wide view of the forcings at stakes on the monsoon phenomenon, while Tada et al. (2007) propose a view of both ISM and SEAM evolution with respect to Himalayan and Tibetan orogeny (Fig 7 in particular). Molnar et al. (2010) also propose a rather comprehensive view of the major monsoon indicators available in Asia since the last 20 Ma. If their evolution is usually not precise up to the orbital time-scale, they should be used to validate and/or discuss the longer time-scale pattern of amplification you are simulating and the remaining uncertainties.

Thanks, we have included Tada (2016) [34] and Molnar (2010) [35] in the proxy discussion in response to R#2 above, and pasted again below.

It is instructive to compare these long-term changes with a review of observational evidence³⁴ that concluded that the SEAM intensified between 25 and 20 Ma, putatively in response to northern HTP uplift. Further intensification has been associated with northeastern and eastern HTP uplift from 15 to 10 Ma^{35,36}, possibly masked by the competing effects of CO₂ drawdown, cooling and increasing glaciation³⁴. Our emulations are broadly consistent with these inferences, although orographic forcing dominates in our results, and clear net strengthening of SEAM rainfall is emulated in response. Proxy evidence further indicates transient increases in SEAM intensity at around 16-14 Ma and 4.2–2.8 Ma³⁴ where orography is dominant in our SEAM emulation. However, we do emulate an intensification in ISM rainfall around these times (14-12 Ma and 4-3 Ma). During the Pleistocene, proxies indicate

SEAM weakening, but with increased orbital variability³⁴. This weakening is apparent (though modest) in the SEAM emulation, driven by a combination of ice-sheet and CO₂ forcing. Although increased variability is evident in our emulated records, this increase occurs earlier, over the period 15-13 Ma, possibly associated with the final HTP uplift, unrelated to Pleistocene boundary conditions. Orbital timescale monsoon variability has been evidenced in proxy records throughout the last 30 Ma and earlier^{37,38,39}, suggestive of both low-latitude precessional forcing and high-latitude obliquity forcing via ice sheets, and consistent with our emulators. A direct comparison of orbital variability between our regional-scale emulators and individual, locally-influenced proxy records would be extremely challenging⁴⁰, but possible in principle if a sufficient density of observational records could be assembled.

As mentioned by R#1 we believe **Botsyun et al. 2019** should be cited as well (Line 176), given they represent the main alternative hypothesis to paleoaltimetry-based methods from Rowley et al. and Spicer et al.

Added as reference 46.

**Additional questions from R#3 to the authors:
On the manuscript**

Line 17: “multiple drivers of Asian monsoon variability1: orbital forcing, atmospheric carbon dioxide (CO₂), global ice volume, Himalayan-Tibetan Plateau (HTP) uplift, tectonically-induced changes to major ocean gateways, and Intertropical Convergence Zone (ITCZ) movement, via atmospheric circulation changes and Atlantic Meridional Ocean Circulation (AMOC) teleconnections » **R#3** You mention here and further analyze the closing of the Tethys, but totally overlook the impact of continentality changes induced by the retreat of the Paratethys, initiating in the Eocene and continuing during the Oligocene and the Miocene (Kaya et al. 2019). Studies have shown that the increased continentality over the progressively emerging Anatolia, Arabia, North Africa and central Europe (induced by the Thetys and Paratethys retreat due to global eustasy and to the convergence of Africa and Eurasia) do trigger changes on the monsoon strength (**Zhang et al. 2007, Ramstein et al. 1997, Fluteau et al. 1999**). It is mostly due to a displacement of main low and high sea level pressure patterns over the continents, that drive increased moisture convergence over Asia. This is also discussed in Tardif et al. 2020. Although testing this would require more refined paleogeography reconstructions, which is out of the scope here, I think it is paramount to mention it, in the introduction and also in the results regarding the Tethys, since it is to me the major (only) point that your study hasn’t considered (line ~133 for ex).

Addressed with added text in introduction (see above)

Line 126:

R#3 I find generally confusing that most conclusions in the text are made considering the annual rainfall changes induced by changing parameters, whereas the corresponding figures display seasonal JJA precipitation changes (Fig. S9, S17). You state that SEAM changes significantly under cross-interaction between forcings (Line 126: from 1320 to 1300 mm and from 1216 to 1242 mm). This represent changes of ~20-26 mm/yr that you find significant. However, when testing the solar constant reduction (see further comment for Line 330), the

change of ~28 mm/yr on the ISM is considered not significant. I understand that, given the wider variability in annual ISM rainfall, this change induced by solar constant might indeed be of second order. However, you use the figures S9 and S17 to support these conclusions, whereas these figures show JJA precipitations anomalies that are both in the order of +/- 1mm in the ISM domain. I find this mixture in seasonal versus annual rainfall blurs the message. I think the Supp Figures should at least display the anomalies mentioned in the text (annual rainfall values). Moreover, you dismiss the effect of the solar constant based on its minimal square root value compared to other values showed in Table 1. This add a third way to explain a single variability. You might pick one that is understandable (square root, annual or seasonal anomaly) or show all.

We chose to emulate annual rainfall as best captured by proxies (runoff etc), but to validate with a focus on JJA climate to highlight the dynamics, in particular to demonstrate that seasonal variability is reasonably simulated. However, we agree this makes comparisons difficult and to address this have added annual precipitation to S16, and have also added a new figure S17 showing annual precipitation change for the five dominant drivers. These figures emphasise the effects of solar constant reduction are second order (noting the legend scales in S16 and S17 differ by an order of magnitude).

We agree that the discussed changes due to the cross-interactions are comparable to the solar constant, while noting these are second-order effects driven by interactions, not the primary effects of a forcing. We also note that these interactions are “instantaneous”, driven by Tethys closure, in contrast to the gradual evolution of the solar constant over 30 Ma. We included this discussion we felt as the step changes in Fig. 2 (now Fig. 3) needed explanation, especially the CO2 sensitivity, which highlights a significant (100 ppm) interaction between CO2 and Tethys closure.

The square root (total effect) is the standard deviation of the annual precipitation change driven by a parameter - to approximate this we simply assumed the anomaly changes from 28 mm to zero over 30 Ma (neglecting interactions) and took the standard deviation. We feel it is necessary to present it in this way for an apples-with-apples comparison with the other variables.

Lines 136, 140:

R#3 You refer to a « compression » of the ITCZ due to ice sheet buildup, that « strengthens the tropical hydrological cycle ». I don't find this statement very clear from a physical or mechanistic point of view: does it mean the precipitations quantities are unchanged but concentrated over a narrower band? that the moisture convergence is increased? that the wind speed increases?

We have revised this to simply read

Both SEAM and ISM rainfall show greater sensitivity to orbital forcing since about 14 Ma, apparently associated with the final phase of HTP uplift.

Line 150: « We infer this ISM rainfall increase to be driven by indirect teleconnections arising from constriction of the Panama gateway, which led to increased AMOC²⁹ (Supplementary Fig.6) and consequent northward movement of the ITCZ. »

R#3 What would be the link between an increased AMOC and a northward migration of the ITCZ? are there modelling studies or proxies allowing to make this assumption?

Thanks. We have added a clause of explanation (ocean heat) and citations. There are a number of studies that have addressed this, and we have chosen to add two relatively recent citations (Frierson et al 2013, Marshall et al 2014) which explore the mechanisms.

We infer this ISM rainfall increase to be driven by indirect teleconnections arising from constriction of the Panama gateway, which led to increased AMOC³¹ (Supplementary Fig. 5), redistribution of interhemispheric ocean heat and consequent northward movement of the ITCZ^{32,33}.

Line 300: “We note that precession was also found to be the dominant driver of Eocene monsoon variability simulated by PLASIM-GENIE in a fully altered paleogeography⁷⁰, thus the details of the geography do not appear to be important » **R#3** This statement is problematic to me because these forcings don't act on the same timescales. Precession is the dominant mode of the monsoon variability on time scales of several thousands of years. On longer timescales, paleogeography becomes the dominant mode. Moreover, as you average your results over large regions and with low spatial resolution, you cannot assume the results would be strictly similar in a finer resolved model. Please consider removing “thus the details of the geography do not appear to be important”

Agreed, we have removed this clause.

Line 330 (Fig S17): « Simulated annual SEAM rainfall is unchanged, ISM annual rainfall is reduced by 28mm, and there are changes in the seasonal distributions of both systems. However, these changes are modest (implied $\sqrt{\text{total effect}} \sim 10\text{mm}$) compared to the other drivers (Table 1). » **R#3** Could you add the anomaly plots on the annual rainfall too? Otherwise, it seems weird to say that SEAM annual rainfall is unchanged but showing in Fig S17 that summer precipitation is significantly increased in this region. Your paper investigates regional summer monsoons, and your figure actually shows an increase in summer SEAM precipitation due to increase solar constant. If you decide to base your arguments regarding monsoon strengthening on the annual rainfall metric (like Farnsworth et al.), then showing this annual anomaly in S17 seems to better serve your purpose.

Thanks, annual rainfall change added as Fig S17c (see above)

Table 1

R#3 : the 3 central columns referring to the Miocene are unclear to me. Is the "Miocene" column somehow a weighted average of the values obtained for Late and early Miocene? Given that the impact of obliquity on the “Miocene” is higher (17) than for the early (15) and the late (16) Miocene, I am confused.

You mention that more details are available in Supp Material 1.2, but this section doesn't appear clearly in the supplementary file I have.

Apologies, the reference to SM1.2 was a relic from the initial submission as a letter. This explanation can be found in the methods, now corrected.

With regard to the apparent inconsistency for obliquity during the Miocene, although the total effects are dominantly a function of the priors of the parameter in question, they are also a function of the priors of the other parameters, that are held fixed. Therefore, even if the priors for the parameter being varied are identical, they will be varied within an ensemble that takes different fixed values for the other inputs and the interactions with these other parameters will lead to different variability. To illustrate, the CO₂ prior has similar variance across the whole Miocene as across the late Miocene, but with a higher mean (see Supp Table 1), and we speculate that this higher CO₂ is driving the slightly higher variance in response to obliquity.

We have added text to clarify in the figure caption.

Note that while the Total Effects are dominantly a function of the prior of the parameter under question, they also depend upon the priors of the other variables through parameter interactions (methods). For this reason, the total effects in the Miocene cannot be derived from the values in the late and early Miocene, and may take values outside of their range.

On the Supplementary Information:

FigS14:

R#3 The legend of the figure should be made clearer. What are the values in y axes (mm I guess)? What is the legend of the color bars (black, orange, yellow), mean, median and 5-95% interval?

Thanks, this is clarified with caption text below and the units have been added to the figure titles.

Orange bars are ensemble medians, yellow bars are the 90% confidence intervals and black bars are the simulations from the optimised parameter set used to build the emulators.

Additional references mentioned in my comments:

Ao, H., Dupont-Nivet, G., Rohling, E. J., Zhang, P., Ladant, J.-B., Roberts, A. P., Licht, A., Liu, Q., Liu, Z., Dekkers, M. J., Coxall, H. K., Jin, Z., Huang, C., Xiao, G., Poulsen, C. J., Barbolini, N., Meijer, N., Sun, Q., Qiang, X., Yao, J. and An, Z.: Orbital climate variability on the northeastern Tibetan Plateau across the Eocene–Oligocene transition, *Nature Communications*, 11(1), doi:10.1038/s41467-020-18824-8, 2020.

Meijer, N., Dupont-Nivet, G., Abels, H. A., Kaya, M. Y., Licht, A., Xiao, M., Zhang, Y., Roperch, P., Poujol, M., Lai, Z. and Guo, Z.: Central Asian moisture modulated by proto-Paratethys Sea incursions since the early Eocene, *Earth and Planetary Science Letters*, 510, 73–84, doi:10.1016/j.epsl.2018.12.031, 2019.

Huang, C. and Hinnov, L.: Astronomically forced climate evolution in a saline lake record of the middle Eocene to Oligocene, Jiangnan Basin, China, *Earth and Planetary Science Letters*, 528, 115846, doi:10.1016/j.epsl.2019.115846, 2019.

Molnar, P., Boos, W. R. and Battisti, D. S.: Orographic Controls on Climate and Paleoclimate of Asia: Thermal and Mechanical Roles for the Tibetan Plateau, *Annual Review of Earth and Planetary Sciences*, 38(1), 77–102, doi:10.1146/annurev-earth-040809-152456, 2010.

Kaya, M. Y., Dupont-Nivet, G., Proust, J., Roperch, P., Bougeois, L., Meijer, N., Frieling, J., Fioroni, C., Özkan Altiner, S., Vardar, E., Barbolini, N., Stoica, M., Aminov, J., Mamtimin, M. and Zhaojie, G.: Paleogene evolution and demise of the proto-Paratethys Sea in Central Asia (Tarim and Tajik basins): Role of intensified tectonic activity at ca. 41 Ma, *Basin Research*, 31(3), 461–486, doi:10.1111/bre.12330, 2019.

Zhang, Z., Wang, H., Guo, Z. and Jiang, D.: What triggers the transition of palaeoenvironmental patterns in China, the Tibetan Plateau uplift or the Paratethys Sea retreat?, *Palaeogeography, Palaeoclimatology, Palaeoecology*, 245(3–4), 317–331, doi:10.1016/j.palaeo.2006.08.003, 2007.

Ramstein, G., Fluteau, F., Besse, J. and Joussaume, S.: Effect of orogeny, plate motion and land–sea distribution on Eurasian climate change over the past 30 million years, *Nature*, 386(6627), 788–795, doi:10.1038/386788a0, 1997.

Fluteau, F., Ramstein, G. and Besse, J.: Simulating the evolution of the Asian and African monsoons during the past 30 Myr using an atmospheric general circulation model, *Journal of Geophysical Research: Atmospheres*, 104(D10), 11995–12018, doi:10.1029/1999JD900048, 1999.

Braconnot, P., Joussaume, S., Marti, O. and de Noblet, N.: Synergistic feedbacks from ocean and vegetation on the African Monsoon response to Mid-Holocene insolation, *Geophysical Research Letters*, 26(16), 2481–2484, doi:10.1029/1999GL006047, 1999.

Liu, X., Kutzbach, J. E., Liu, Z., An, Z. and Li, L.: The Tibetan Plateau as amplifier of orbital-scale variability of the East Asian monsoon: TIBET PLATEAU AND MONSOON VARIABILITY, *Geophysical Research Letters*, 30(16), doi:10.1029/2003GL017510, 2003.

Reviewers' Comments:

Reviewer #2:

Remarks to the Author:

This is my third review of the manuscript. My main point from the previous round on adding the connection to existing proxy-based reconstructions is included now. This is a very good addition making the paper more complete. The response of the authors to the comments by the other reviewer have also extensively been answered. So I think the manuscript is ready now for publication. And concerning the attention especially the Miocene and monsoons in general receive this paper will be a valuable contribution to Nat. Comm. Below are a few minor comments.

Something is not smoothly running when going from the first to the second sentence in the new proxy paragraph.

"Proxy evidence further indicates transient increases in SEAM intensity at around 16-14 Ma and 4.2-2.8 Ma where orography is dominant in our SEAM emulation. However, we do emulate an intensification in ISM rainfall around these times (14-12 Ma and 4-3 Ma)."

I would re-word "observations/observational" to something like reconstructed.

Page 4: "Both SEAM and ISM rainfall show greater sensitivity to orbital forcing since about 14 Ma, apparently associated with the final phase of HTP uplift." Still, this is also a main glaciation phase on Antarctica (MMCT) that would increase orbital forcing from high southern latitudes (northern hemisphere ice sheets only start to play their role in the Pliocene), i.e. obliquity. This could be interpreted as the HTP uplift causing this glaciation, or?

I would change the headings India and SE Asia in Table 1 to ISM and EASM as everywhere else in the text.

Reviewer #3:

Remarks to the Author:

Dear authors,

First, my apologies for posting this review slightly after the deadline.

I feel all of my comments have been addressed, apart from the one below :

R#3: Lines 136, 140: You refer to a « compression » of the ITCZ due to ice sheet buildup, that « strengthens the tropical hydrological cycle ». I don't find this statement very clear from a physical or mechanistic point of view: does it mean the precipitations quantities are unchanged but concentrated over a narrower band? that the moisture convergence is increased? that the wind speed increases?

Authors: We have revised this to simply read

"Both SEAM and ISM rainfall show greater sensitivity to orbital forcing since about 14 Ma, apparently associated with the final phase of HTP uplift."

I feel the answer doesn't address my question. I still suggest the authors should clarify the statement "ITCZ compression and increased tropical hydrological intensity". Moreover, I don't think this new statement is exact, since your results (Table 1) clearly show that only SEAM display greater sensitivity to orbital forcing since the Pliocene, while ISM display a strong sensitivity to orbital forcing in all tested stages.

Apart from this point, I think this paper is ready to be published and I thank the authors for their comprehensive answers.

Best Regards
Delphine Tardif

Response to reviewers, Asian monsoon drivers, Thomson et al

We are grateful for the continued efforts of both reviewers. Our responses are given below in red, with edits to the manuscript in green.

Reviewer #2 (Remarks to the Author):

This is my third review of the manuscript. My main point from the previous round on adding the connection to existing proxy-based reconstructions is included now. This is a very good addition making the paper more complete. The response of the authors to the comments by the other reviewer have also extensively been answered. So I think the manuscript is ready now for publication. And concerning the attention especially the Miocene and monsoons in general receive this paper will be a valuable contribution to Nat. Comm. Below are a few minor comments.

Something is not smoothly running when going from the first to the second sentence in the new proxy paragraph. "Proxy evidence further indicates transient increases in SEAM intensity at around 16-14 Ma and 4.2–2.8 Ma where orography is dominant in our SEAM emulation. However, we do emulate an intensification in ISM rainfall around these times (14-12 Ma and 4-3 Ma)."

We have rephrased to:

Proxy evidence further indicates transient increases in SEAM intensity at around 16-14 Ma and 4.2–2.8 Ma³⁴. Similar events are seen in the ISM emulator at around these times (14-13 Ma and 4-3 Ma), but are absent in the SEAM emulator, which is dominated by orography in this period.

I would re-word "observations/observational" to something like reconstructed.

Changed throughout to proxy reconstructions/reconstructions.

Page 4: "Both SEAM and ISM rainfall show greater sensitivity to orbital forcing since about 14 Ma, apparently associated with the final phase of HTP uplift." Still, this is also a main glaciation phase on Antarctica (MMCT) that would increase orbital forcing from high southern latitudes (northern hemisphere ice sheets only start to play their role in the Pliocene), i.e. obliquity. This could be interpreted as the HTP uplift causing this glaciation, or?

Changed to

Both SEAM and ISM rainfall show increasing sensitivity to orbital forcing from about 15 Ma, peaking by around 12 Ma, associated with times of increasing Antarctic glaciation and orogeny forcing (Figure 2).

I would change the headings India and SE Asia in Table 1 to ISM and EASM as everywhere else in the text.

Done

Reviewer #3 (Remarks to the Author):

Dear authors,

First, my apologies for posting this review slightly after the deadline.

I feel all of my comments have been addressed, apart from the one below :

R#3: Lines 136, 140: You refer to a « compression » of the ITCZ due to ice sheet buildup, that « strengthens the tropical hydrological cycle ». I don't find this statement very clear from a physical or mechanistic point of view: does it mean the precipitations quantities are unchanged but concentrated over a narrower band? that the moisture convergence is increased? that the wind speed increases?

Authors: We have revised this to simply read

"Both SEAM and ISM rainfall show greater sensitivity to orbital forcing since about 14 Ma, apparently associated with the final phase of HTP uplift."

I feel the answer doesn't address my question. I still suggest the authors should clarify the statement "ITCZ compression and increased tropical hydrological intensity".

Thank you for pressing this point, which motivated us to investigate the mechanism in more detail and to conduct a further ensemble analysis to investigate robustness (Δ ICE added to a revised Supplementary Fig 13) reproduced below.

Supplementary Fig.13: Perturbed parameter ensembles using the 69-member pre-calibrated parameter set of Holden et al (2018). These ensembles considered the dominant forcings (Table 1) of orogeny (scaling by 50%), CO₂ (doubling to 560ppm), precession (reversed phase) and LGM ice sheets. Orange bars are ensemble medians, yellow bars are the 90% confidence intervals and black crosses are the simulations from the optimised parameter set used to build the emulators.

The ice-sheet sensitivity simulation shown in Supplementary Fig 4 reveals a weakened JJA Hadley circulation that drives reduced JJA precipitation (Supplementary Fig 2). This reduction in JJA precipitation is robust across the new ensemble (cf ensemble mean below), but is offset by an increase in DJF precipitation over India that results from a strengthening of winds blowing onto the subcontinent from the Bay of Bengal. The relative contributions of these seasonal changes to the annual rainfall are not robust (Supplementary figure 14 above).

To address this we have revised the text in three places.

1) Discussion of ice-sheet related mechanisms removed (Paragraph 4 of *Monsoon rainfall changes since 30Ma*). Text now reads:

Furthermore, CO₂ drawdown began after 15 Ma and continued more-or-less until the Industrial Revolution (Fig. 1), driven by HTP weathering, and led to global cooling²⁹. These three effects – atmospheric disturbance caused by orogeny, CO₂ drawdown, and increasing ice volume – produced competing effects in terms of monsoon rainfall. Both SEAM and ISM rainfall show increasing sensitivity to orbital forcing from about 15 Ma, peaking by around 12 Ma, associated with increasing Antarctic glaciation and orogeny forcing (Figure 2).

2) Mechanism described here, together with discussion of robustness (Paragraph 6 of *Monsoon rainfall changes since 30Ma*).

Falling CO₂ after 3 Ma drove a reduction in ISM rainfall (Fig. 3f), despite a competing increase driven by Pleistocene glaciation. The ice-sheet sensitivity simulation reveals a weakened JJA Hadley circulation (Supplementary Fig 4) that drives reduced JJA precipitation (Supplementary Fig 2), but this is offset in the annual average by an increase in DJF precipitation due to strengthened winds blowing onto the subcontinent from the Bay of Bengal. The relative contributions of these seasonal changes are not robust under parametric uncertainty (Supplementary Fig. 13) and should be treated with caution.

3) Methods text to cover the new ensemble analysis (see *PLASIM-GENIE parametric uncertainty*). The new text is in boldface (in this response only)

Model uncertainty is not captured by the emulator, because it was built from simulations with a single parameter set. In order to validate the robustness of the results under model uncertainty, we performed a series of perturbed parameter ensembles using a 69-member pre-calibrated parameter set⁷³. These ensembles considered the forcings of orogeny (scaling by 50%), CO₂ (doubling to 560 ppm), precession (reversed phase) **and LGM ice sheets** and are summarised in Supplementary Fig. 13. Under parametric uncertainty, the three dominant drivers of variability (Table 1) namely orogeny, CO₂ and precession, all drive ISM change that is significant with respect to the baseline preindustrial state and, despite significant uncertainty, all drive change of consistent sign. In all simulations, both doubled CO₂ and precession-reversal drive a strengthening of the ISM while reduced orogeny drives a weakening. The optimised parameter set used to build the emulator lies close to the centre of all three ensembles. **In contrast, the negligible role of ice sheets (Table 1, Fig. 3) on the ISM is not robust under parametric uncertainty and should be treated with caution.** Consistent with the emulator, simulated SEAM rainfall is dominated by orogeny in all ensemble members. SEAM changes driven by CO₂ and precession are modest and (in contrast to the ISM) of uncertain sign, indicating that the phasing of orbitally-driven SEAM change should be treated with caution. We note that precession was found to be the dominant driver of Eocene monsoon variability simulated by PLASIM-GENIE in a fully altered paleogeography⁸². **Reduced SEAM rainfall in response to LGM ice sheets is robust under parametric uncertainty.**

Moreover, I don't think this new statement is exact, ["Both SEAM and ISM rainfall show greater sensitivity to orbital forcing since about 14 Ma, apparently associated with the final phase of HTP uplift."] since your results (Table 1) clearly show that only SEAM display greater sensitivity to orbital forcing since the Pliocene, while ISM display a strong sensitivity to orbital forcing in all tested stages.

We believe this statement is correct, noting edits in response to reviewer #2 above: Both SEAM and ISM rainfall show increasing sensitivity to orbital forcing from about 15 Ma, peaking by around 12 Ma, and apparently associated with increasing orogeny forcing. **To clarify, from Table 1 (pasted below for convenience with precession in red font), ISM sensitivity to precession is 55 to 59 mm in the Oligocene and early Miocene, 91 to 127 mm in Late Miocene and later. SEAM sensitivity to precession is 15 to 17mm in the Oligocene and early Miocene, 30 to 39 mm in Late Miocene and later. While we agree that there is significant ISM sensitivity in all stages (dark green shading), it is also true that the sensitivity increases (from ~50 mm to ~100 mm) from the mid Miocene as stated. Note this is also apparent from Fig. 3g.**

	Holo-cene	Pleisto-cene	Pliocene	Miocene	Late Miocene 15.97- 5.333Ma	Early Miocene 23.03- 15.97Ma	Oligo-cene	30Ma to pre-industrial	30Ma to pre-industrial % of total
ISM									
CO ₂	13 ± 2	44 ± 6	63 ± 11	55 ± 13	73 ± 4	14 ± 1	16 ± 1	71 ± 5 21%	
sea level	9 ± 1	16 ± 1	14 ± 1	31 ± 3	28 ± 3	29 ± 1	24 ± 1	35 ± 3 5%	
precession	127 ± 4	106 ± 5	123 ± 4	91 ± 5	91 ± 8	59 ± 3	55 ± 5	77 ± 5 25%	
obliquity	11 ± 2	9 ± 1	12 ± 1	20 ± 1	18 ± 1	17 ± 1	13 ± 1	17 ± 1 1%	

eccentricity	43 ± 1	36 ± 1	44 ± 0.4	32 ± 2	33 ± 3	25 ± 1	25 ± 1	30 ± 2	4%
orogeny	0	0	2 ± 0.1	65 ± 4	66 ± 4	0	20 ± 1	97 ± 3	39%
gateways	-	-	-	-	-	-	-	35 ± 1	5%
Total	135	121	146	134	142	74	71	155	100%
SEAM									
CO ₂	2 ± 0.1	20 ± 2	11 ± 0.4	23 ± 1	24 ± 1	6 ± 1	11 ± 0.3	31 ± 1	1%
sea level	9 ± 0.3	16 ± 0.2	13 ± 1	31 ± 1	33 ± 1	24 ± 1	24 ± 0.2	35 ± 1	1%
precession	39 ± 3	36 ± 3	38 ± 2	31 ± 1	30 ± 1	17 ± 1	15 ± 0.4	28 ± 1	1%
obliquity	14 ± 1	13 ± 1	10 ± 0.4	10 ± 1	11 ± 1	9 ± 1	8 ± 1	10 ± 1	<1%
eccentricity	19 ± 4	18 ± 2	19 ± 1	20 ± 1	19 ± 1	14 ± 0.3	12 ± 1	17 ± 1	<1%
orogeny	0	0	5 ± 1	184 ± 1	181 ± 1	0	109 ± 2	298 ± 1	93%
gateways	-	-	-	-	-	-	-	55 ± 1	3%
Total	46	49	47	192	189	34	114	309	100%

Apart from this point, I think this paper is ready to be published and I thank the authors for their comprehensive answers.

Best Regards

Delphine Tardif

Reviewers' Comments:

Reviewer #3:

Remarks to the Author:

I thank the authors for their insightful additional descriptions regarding LGM Ice sheets effect on ISM and SEAM.

I believe all my comments were answered and doubts clarified.

One last thing I'd like to point out is the somewhat odd legends from most Figures in the SI, where tick marks don't match increment colors transitions. I trust the authors would agree that the concerned figures would surely benefit from a more straightforward legend, but I'll leave it to their appreciation, given I should have mentioned it in the first round of review.

I think this paper will improve greatly our understanding of the main forcings constraining the Asian monsoons evolution.

Response to reviewers, Asian monsoon drivers, Thomson et al

We thank the reviewer for their careful consideration and constructive criticism throughout this process which has greatly strengthened the paper. In regard to the legend tick marks in the SI (see below), these were a deliberate choice. We preferred to use a large number of colours (typically 11-17) in order to maximise the information conveyed and, for anomaly plots, we prefer an odd number of colours centred on white to clearly delineate the sign transition. These choices have the consequence that tick marks do not line up with colour transitions. While we could change this by e.g. moving to an even number of colours, fewer colours or a graded colour scale, we feel that each of these choices have their own disadvantages and the considerable amount of work involved is not justified, especially not for an essentially subjective and aesthetic preference. We note that there is no measurable impediment to the scientific understanding of the figures and thank the reviewer for leaving this decision to us.

Best wishes, Phil Holden on behalf of all authors

Reviewer #3 (Remarks to the Author):

I thank the authors for their insightful additional descriptions regarding LGM Ice sheets effect on ISM and SEAM.

I believe all my comments were answered and doubts clarified.

One last thing I'd like to point out is the somewhat odd legends from most Figures in the SI, where tick marks don't match increment colors transitions. I trust the authors would agree that the concerned figures would surely benefit from a more straightforward legend, but I'll leave it to their appreciation, given I should have mentioned it in the first round of review.

I think this paper will improve greatly our understanding of the main forcings constraining the Asian monsoons evolution.